# Fracture Seismic: Mapping Subsurface Connectivity

**Charles Sicking [1,*] and Peter Malin [2]**

[1]  Ambient Reservoir Monitoring, 3701 Anatole Ct., Plano, TX 75075, USA

[2]  Advanced Seismic Instrumentation & Research, 1311 Waterside, Dallas, TX 75218-4475, USA; pem@asirseismic.com

*  Correspondence: charles@ambientreservoir.com; Tel.: 1-214-763-6711

**Abstract:** Fracture seismic is the method for recording and analyzing passive seismic data for mapping the fractures in the subsurface. Fracture seismic is able to map the fractures because of two types of mechanical actions in the fractures. First, in cohesive rock, fractures can emit short duration energy pulses when growing at their tips through opening and shearing. The industrial practice of recording and analyzing these short duration events is commonly called micro-seismic. Second, coupled rock–fracture–fluid interactions take place during earth deformations and this generates signals unique to the fracture's physical characteristics. This signal appears as harmonic resonance of the entire, fluid-filled fracture. These signals can be initiated by both external and internal changes in local pressure, e.g., a passing seismic wave, tectonic deformations, and injection during a hydraulic well treatment. Fracture seismic is used to map the location, spatial extent, and physical characteristics of fractures. The strongest fracture seismic signals come from connected fluid-pathways. Fracture seismic observations recorded before, during, and after hydraulic stimulations show that such treatments primarily open pre-existing fractures and weak zones in the rocks. Time-lapse fracture seismic methods map the flow of fluids in the rocks and reveal how the reservoir connectivity changes over time. We present examples that support these findings and suggest that the fracture seismic method should become an important exploration, reservoir management, production, and civil safety tool for the subsurface energy industry.

**Keywords:** fracture seismic; fracture connectivity; fracture mapping; passive seismic;

## 1. Introduction

We use the term "Fracture Seismic Method" to refer to the method of mapping fractures using one-way depth migration applied to fracture emissions that have durations of seconds to minutes. The use of the term in this fashion distinguishes the fracture seismic method from other methods such as the reflection seismic method and the micro-seismic method. This paper presents an end-to-end description of the fracture seismic method and presents examples that map subsurface connectivity structures. The fracture seismic method extends current passive methods by making use of harmonic resonances within the fracture that are caused by interfering Krauklis waves (Krauklis, 1962) [1] initiated by dislocations on fracture tips and internal fracture fluid flows (e.g., Frehner, 2014 [2]; Tary et al., 2014) [3,4]. The fracture emissions come from short duration energy pulses and harmonic resonances of the entire fracture. The resonances are episodic, seconds to minutes long, and occur in the frequency band of 1 to 100 Hz. They are readily observed in passive, multichannel seismic recordings at both green- and brown-field sites. Two examples of fracture seismic signals are shown in the spectrograms in Figure 1.

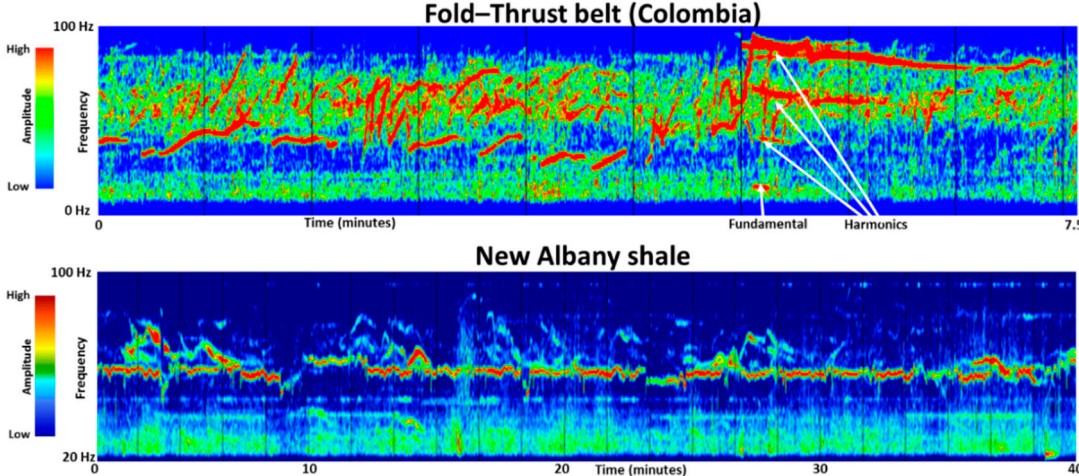

**Figure 1.** Spectrograms of fracture seismic data containing resonances. The top panel is from a Colombia thrust zone where the regional compressional stress is high. In the first 5 min of this panel, there are two styles of resonances. Note the harmonics at 5 min. The bottom panel is from the New Albany shale. It reveals a much simpler resonance signal where the highest intensity resonance is in the 50 to 60 Hz frequencies, with lower intensities at lower frequencies. Figure modified from Sicking et al. (2019) [5,6].

The most widely used method for monitoring of hydraulic fracturing uses geophones at reservoir depth in vertical wells that are located near the hydraulic fracturing. Maxwell et al. (2003) [7] describe this downhole method for detecting microearthquakes (MEQ) generated during stimulation operations and for imaging deformation associated with the injections.

Another method for mapping MEQ during the hydraulic fracturing uses surface or buried grid recordings. The basis of this method is Kirchhoff migration, and it is normally referred to as seismic emission tomography. Duncan et al. (2010) [8] describe the surface geophone method for detecting and mapping MEQ. The focus of these hydraulic fracture monitoring methods is to use the MEQs to infer the creation of fracture connectivity.

Kochnev et al. (2007) [9] describe a non-MEQ passive seismic imaging method for mapping the progression of hydraulic fracturing that is similar to the fracture seismic depth migration method presented here. Their method requires searching the trace data for low-energy source seismic waves that can be identified before imaging and the method is applied only to map the progression of the stimulation over time. This approach is not useful for mapping pre-existing fractures before drilling.

In work related to our fracture seismic method, Tary et al. (2012) [10] compute continuous time-frequency transforms that highlight signals that have time-varying resonance frequencies. They conclude that these signals are the result of resonance in fluid-filled fractures or, alternatively, successions of very small repetitive seismic events along the fractures. They also observe correlations between the variations in the frequency content of their recordings, the hydraulic fracturing conditions, and the occurrence of micro-seismic events. They note that there is a direct correspondence between variations in the slurry injection rate and the combined energy emitted.

Seeking to better identify these ambient emissions as opposed to MEQ events, Chorney et al. (2012) [11] present results on seismic energy sources that are associated with deformations such as tensile fracturing or slow slips. Furthermore, Bame et al., (1986) [12] note that the ambient signals they observe are unlikely to be detected by searching with seismic event triggering methods because these require sharp signal onsets.

Additional support for the origins of these episodic signals that occur over long time intervals can be found in the fracture mechanics literature. Vermilye et al. (1998) [13] and Shipton et al. (2001) [14] investigate the various release mechanics of stored elastic strain energy from rocks through field studies of fractures. This stored strain energy is not evenly distributed in the earth's crust, but it is

preferentially released on fracture/fault surfaces and in the damage zones surrounding these fractures.

Fracture mechanics theory predicts that stress concentrations are associated with fractures. Accordingly, Vermilye et al. (1998) [13] and Moore et al. (1995) [15] report field and laboratory studies with clear evidence that these stress concentrations are recorded in the fracture damage zones. Vermilye et al. (1998) [13] show that damage zones consist of rock volumes with a high density of small fractures and that the density of fractures increases exponentially with their proximity to the main fracture surface.

Ziv et al. (2000) [16] show that the brittle crust is in a state of unstable frictional equilibrium. Therefore, very small changes in stress (approximately 0.01 bar or approximately 1 kPa) can cause slippage on weak fractures. Lawn et al. (1975) [17] show that failure occurs preferentially on small, optimally oriented fractures and in the zones surrounding the fractures in which crack-tip stress concentrations amplify the stress magnitudes. Hubbert et al. (1959) [18] show that this unstable equilibrium is further disturbed as additional fluid pressures reduce the normal stress on preexisting fractures. They also show that, during production, subtle movement of fluid produces similar effects.

Fracture seismic connectivity mapping started circa 2005. (Geiser et al. 2006) [19]. The end-to-end system for applying the fracture seismic method has been in practice since 2010 (Sicking et al. 2012) [20]. The main mapping step is a time-progressing depth migration of the fracture seismic resonance episodes, a process that we call streaming depth imaging (SDI; Sicking et al., 2016) [21]. Two examples of fracture information that can be computed using SDI are shown in Figure 2. For these examples, the fractures seismic intensity is summed over the time interval of a stimulation stage and the fracture surfaces are extracted from the intensity volume. The left panel shows the fracture surfaces colored by the summed intensity for the entire stage and shows that the summed intensity is highest at the perf shots and lowest at the fractures more distant from the perforations. The right panel shows the fracture surfaces colored by the clock time of the first fracture emissions. The fractures to the left of the well were stimulated first, early in the stage treatment time, and the fractures to the right of the well were stimulated progressively later in time.

Recent research on the source of fracture seismic signals has put the fracture seismic method on a solid theoretical and practical base (e.g., Tary et al., 2014 [3]; Liang et al., 2017) [22]. It has now been applied to dozens of field projects and the examples presented here come from those projects (e.g., Sicking et al., 2014; 2015; 2016; 2017 [21,23–25], Geiser et al., 2012 [26], Lacazette et al., 2013 [27]).

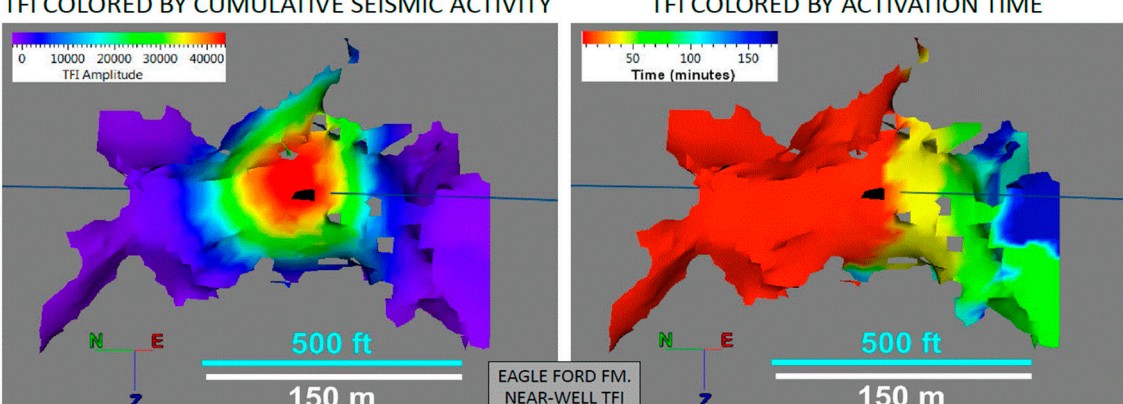

**Figure 2.** Fractures extracted from the local fracture seismic intensity cloud for a single stimulation stage. The left panel shows the extracted fractures colored by the fracture seismic intensity. The intensity at the perf locations (red) are the highest because they are active the longest. The right panel shows the extracted fractures colored by the time of first emission. This shows that the fractures to the left of the well stimulated much earlier during the treatment and the fractures to the right side of the well were stimulated progressively later in time. (Figure from Sicking et.al., 2015) [24].

Several features distinguish fracture seismic from micro-seismic. Micro-seismic uses only slip events that are short enough to allow time separated P and S phases to be recognized on ordinary seismograms (e.g., Aki and Richards, 1980 [28]). Fracture seismic uses signals that can be viewed as the harmonic modes of fluid filled fractures embedded in the upper crust (Liang et al, 2017) [22]. After the resonances are initiated and while there is a continued input of energy, these fracture-length-and aperture-controlled modes of fluid-filled fractures can continue resonating for many seconds to even minutes (Sicking et al., 2019 [5,6]). These waves can also be initiated by the passage of earthquake waves, tectonic and tidal strains, and pressure changes caused by industrial activities. When the geometry of the fracture changes, the frequencies and intensities of the fracture's resonance modes also change.

The methods for observing fracture seismic have been greatly improved by the increase in numbers and sophistication of portable and borehole seismographs over the past few decades. The most cost-effective method is to piggyback on 3D seismic reflection surveys, the fracture seismic data being gathered during active source downtime. The offsets to be covered by the receivers is determined by the target depth of the deepest target and the noise environment. The density of receiving points needed is on the same order as used for recording reflection seismic.

The increased sophistication and speed of seismic reflection signal processing has also significantly aided fracture seismic processing. Initially, many hours of continuous fracture seismic recordings were thought to be necessary in order to build up a 3D volume of fracture intensity. Now, using spectrograms, episodes of intense resonance can be quickly identified and directed into the fracture seismic SDI workflow (Sicking et al. 2019 [5,6]). Many codes used for two-way-travel-time data analysis can be adapted for fracture seismic one-way-travel-time processing. Noise suppression methods are critical in optimal fracture seismic fracture mapping.

After creating a fracture seismic intensity volume via SDI, the local maximum energy surfaces can be tracked and mapped into a 3D image of the connectivity structure. Time-lapse versions of these structures are effective tools for resource management. Because the most intense resonances come from the most permeable fluid filled fractures, changes in relative intensity documents changes in the connectivity and fluid content.

## 2. Background and Methods

### 2.1. Fracture Seismic: Spectrograms

The presence of resonating signals in passive recordings has a long history in observational seismology. They gained prominence in Western literature in connection with volcanic activity (e.g., Dibble, 1972 [29]) and their association with fluid-filled fractures (Aki et al., 1977 [30]). Their occurrence in hydraulic fracturing was inferred during engineered geothermal system studies at Fenton Hills New Mexico (Bame and Felher, 1986 [12]). They have recently been demonstrated to be present in seismic observations of oil and gas stimulations (e.g., Tary et al., 2014[3]). They have now also been identified in quiet time fracture seismic recordings (Sicking et al., 2019[6]). They are most readily seen in time-verses-frequency spectrograms of multichannel seismic data.

Figure 1 shows examples of spectrograms computed from fracture seismic recordings in two very different basins. The top panel is from a thrust zone in Colombia that is under high stress from compressional tectonics, accounting for the high fracture seismic signal level. The first 5 min of this panel show a combination of chaotic and dispersive (frequency changing) resonances. The post 5 min interval shows a resonance with three harmonics.

The bottom panel of Figure 1 shows the spectrogram for data recorded in the New Albany shale during a time before the observation site was hydraulically stimulated. The type of resonances detected during this recording are typical for times when there is no industrial activity. The resonance is dominated by amplitudes in the 50Hz to 60 Hz range, with several lower-intensity bands at lower frequencies. The difference from the Colombian thrust zone is likely due to differences in the state of stress and local geology, which is dominantly extensional.

*2.2. Fracture Seismic: Signal Initiation*

Fracture seismic resonance signals can be initiated in several ways, by both external and internal influences. Examples include distant earthquake strains wave, abrupt responses to accumulated earth tides, isostatic and tectonic deformations, fault creep, and hydraulic stimulation: All contribute energy that can initiate and sustain fracture seismic resonances (e.g., Gomberg, 1996 [31]; Du et al., 2003 [32]; Thomas et al., 2009 [33]; Tary et al., 2014 [3,4]; Liang et al., 2017) [22].

If a fluid-filled fracture is growing, the opening and shearing can initiate the Krauklis waves on the fracture surfaces and they are influenced by the fracture fluid and the surrounding rock. The waves travel along the fracture surfaces, quickly interfering to produce a modal/harmonic resonance of the whole fluid, fracture surface, and surrounding rock system.

Reservoir stimulation by hydraulic fracturing, flooding, or the extraction of fluids causes turbulent flow in the fractures. These flows can also initiate interfering Krauklis waves that then radiate seismic waves. These harmonic motions were first recognized in volcanic activity (e.g., Aki et al., 1977 [30]), next along plate boundaries (e.g., Zhang et al., 2010 [34]), and now in the entire fractured crust (e.g., Sicking et al., 2019 [5,6]).

Tary et al. (2014 [3,4]), divide the resonances into two end-member source types. One type involves unconnected/isolated fractures, the other connected fractures. Their resonant frequencies change with changes in the apertures and lengths of the cracks, and the forces exciting them.

In this paper, examples of resonance and turbulent flow are taken from several different areas and illustrate key features of the spectrograms that are computed for data evaluation and time window selection. Figure 2 show the system of fractures for one stage of stimulation that was computed using fracture seismic data. Both panels show the same fractures, but the left panel fractures are colored by the total local fracture seismic intensity density over the entire stimulation stage and the right panel fractures are colored by the time that each fracture first emits energy.

*2.3. Fracture Seismic: Data Acquisition*

Acquiring fracture seismic data requires a ground surface recording grid of geophones very similar to that used for recording 3D reflection seismic data. The area covered by the receivers should be larger than the area to be imaged. This area is selected such that the edges of the receiver grid are outside of the image area by 1.0 to 1.5 times its depth. If the receivers are laid out on the ground surface, 30 to 60 receiver points are required for each square km. The receivers used for fracture seismic recording on the surface are the same as those used for reflection seismic recording and can record a useful bandwidth that is typically 6 Hz to 1000 Hz. However, in most cases, the recording systems sample the signals at 2 milliseconds and have a Nyquist of 250 Hz. For buried grids where the receivers are placed in boreholes drilled past the local weathering layer, one to three receiver points are required per square km and the receivers may capture frequencies as low as 1 to 2 Hz.

Figure 3 shows four possible layouts for receiver grids. A uniform, face-centered hexagonal distribution is the best design and will have the smallest amplitude artifacts in the final fracture seismic intensity volume. Covering the same area but using cables in an orthogonal grid will provide very good results with only small artifacts in the amplitudes of the fracture seismic intensity volume. The star grid is widely used because it is cheaper to implement in the field and is very versatile in modifying the design to account for access to land. However, the star design is good only for the central portion of the receiver grid. As you move towards the edge of the receiver array, there will be amplitude artifacts and distortions in the locations in the fracture seismic intensity volume. The right side of Figure 3 shows a patch design. This design is sometimes used for projects where only the MEQ detections are desired. However, the patch design causes severe amplitude artifacts in the fracture seismic intensity volumes computed for fracture extractions.

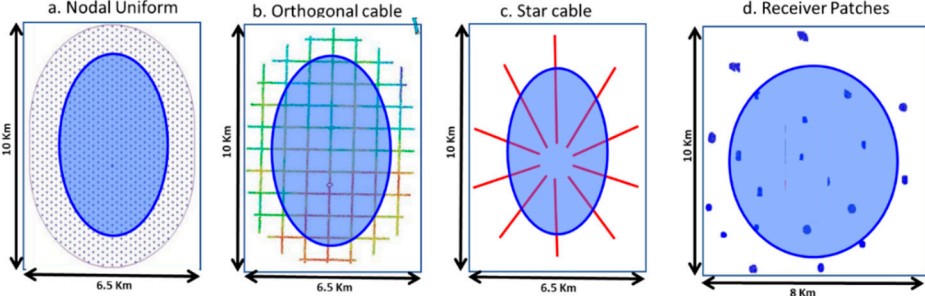

**Figure 3.** Surface recording grids for fracture seismic during the monitoring of stimulations. The grid should cover the desired area around the wells being monitored plus additional area to capture the required aperture for the one-way depth migration. The ideal grid is uniform distribution of geophones as shown in (**a**). The uniform layout has the minimum amount of amplitude artifacts in the fracture seismic intensity volumes. When using a cable system, the geophones are best configured in orthogonal lines (**b**). The star cable layout (**c**) provides good imaging in the center portion of the grid but the fracture seismic intensity volume suffers location distortions in the outer areas. The patch grid (**d**) provides the lowest-quality fracture seismic intensity volumes and has very high amplitude artifacts. Figure modified from Sicking et al. (2019) [5,6].

The depth of a buried grid should be sufficient to place the geophones below the seismic-signal distorting layers of the near surface. The advantage of this type is that the geophones see very little of the surface wave noise that is a major source of interference for surface geophone. For this reason, the density can be reduced to 3 or fewer per square km instead of the 30 to 60 per square km required for surface geophone arrays. Figure 4 shows a buried grid layout in which the density of receivers is 0.45 per square km.

The cost per station is significantly higher for the buried grid than for surface recordings. However, there a many fewer geophone locations and the reduction of noise and the reuse of the same grid for stimulation and time-lapse monitoring makes up for the extra cost. If the geophone grid is reused three times, the cost of the buried grid is less than the cost of the surface geophone grid that is laid out special purpose for each observation. In addition, the quality of fracture seismic maps from a buried grid are much improved over surface grids.

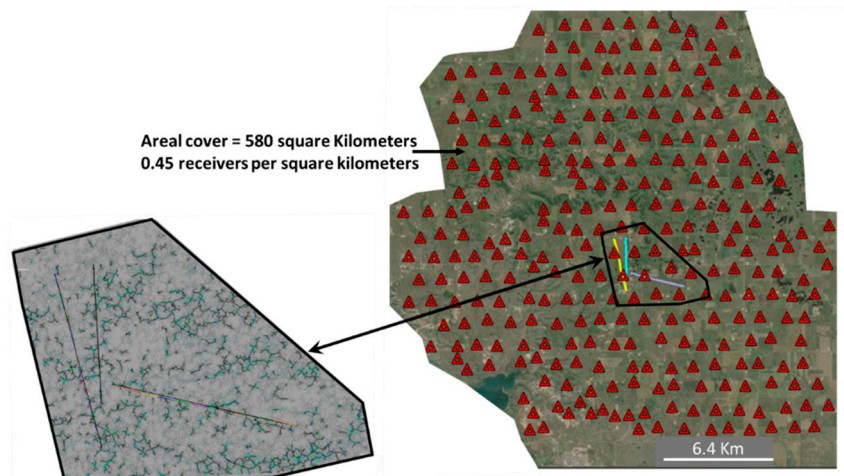

**Figure 4.** Buried geophone arrays are the best option if monitoring will be carried out multiple times over the same area. They are buried 30 to 100 m deep and have the advantage that they do not record the surface wave noise that is encountered on the surface arrays, so the density of geophones is reduced. Surface arrays require 30 to 60 geophones per square kilometer while buried geophone arrays require only 1 to 3 per square km. The geophones can be monitored for each new project by hooking up recorders to each geophone for the time of the project. The figure to the left is a map of the fractures extracted from the survey. Figure modified from Sicking et al. (2019) [5,6].

For the purpose of recording fracture seismic before drilling wells, the data can be collected during the acquisition of 3D reflection seismic, whereby the recorders are switched to continuous recording mode for a few hours while the active sources are offline (Figure 5). The fracture seismic data are recorded at different locations in a roll-along mode as the reflection survey proceeds across the area. Each ground array is recorded and processed separately. The final volumes computed for each array are merged to cover a targeted area.

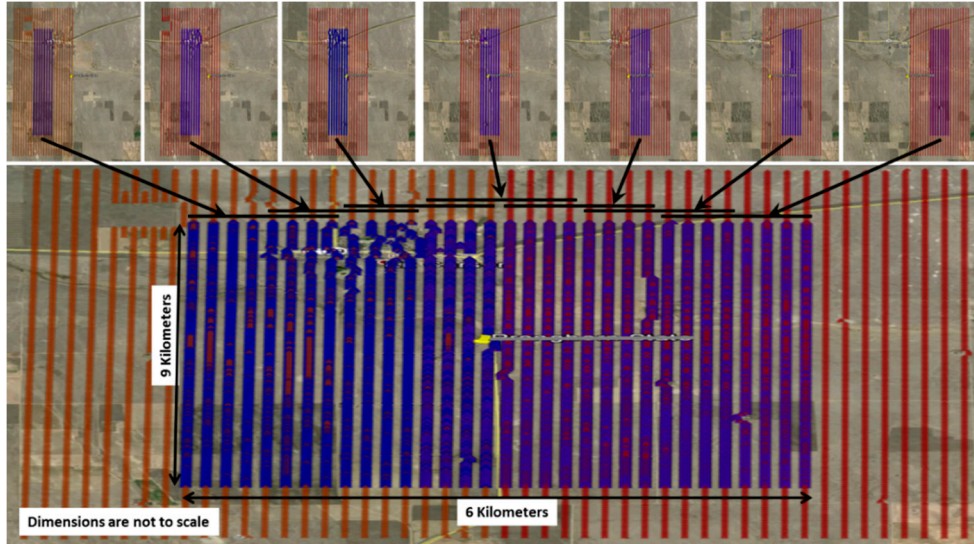

**Figure 5.** Passive seismic recorded using the geophones layout for the 3D reflection seismic recording. The receiver grid is rolled with the 3D acquisition and every few days the active sources are shut down for a few hours while the geophone outputs are recorded in continuous mode. In this example, the area of interest (blue) is recorded in seven separate recording times on different days. The seven fracture seismic intensity volumes have 50% overlap and are merged after the seven final intensity volumes are computed. Merging seven independently recorded and processed volumes causes artifacts at the seams. The fracture seismic intensity volume is discussed in section 3.5.

In this method, however, because overlapping volumes are recorded at different times, the separate volumes see different amounts of fracture seismic energy and the seaming of the volumes can be problematic. The seaming problem can be avoided by laying out geophones over a large surface area and recording continuously for a few days without moving the receivers. An example of this method, discussed later in this paper, is a recording grid that had 4650 active receivers covering a 50 square km study area. In this example, the entire area is recorded simultaneously and only one fracture seismic intensity volume is computed.

The distortions in the intensity volume for different receiver layouts for fracture seismic will differ for different grid configurations. Two examples of amplitude distortions caused by recording grid layout are shown in Figures 6 and 7. Figure 6 shows the amplitude distortions for the patch grid example shown in Figure 3 that has dozens of geophones clustered in each of a few dozen patches. For fracture intensity mapping, the patch design is very poor because of the severe amplitude artifacts in the final volume. The amplitude artifacts are both short and long wavelength and significantly interfere with the interpretation of the fracture system extracted from the recordings.

Figure 7 shows the difference between using a star grid and a cable grid for the same study site. The fracture seismic intensity volume using a star grid does not suppress the highway noise in the final fracture seismic intensity volume while the orthogonal cable grid suppresses the highway noise. The highway is perpendicular to the cables in the star grid and the noise hitting the cable broadside cannot be suppressed. The orthogonal grid suppressed the highway noise because it has cables that are basically parallel to the highway.

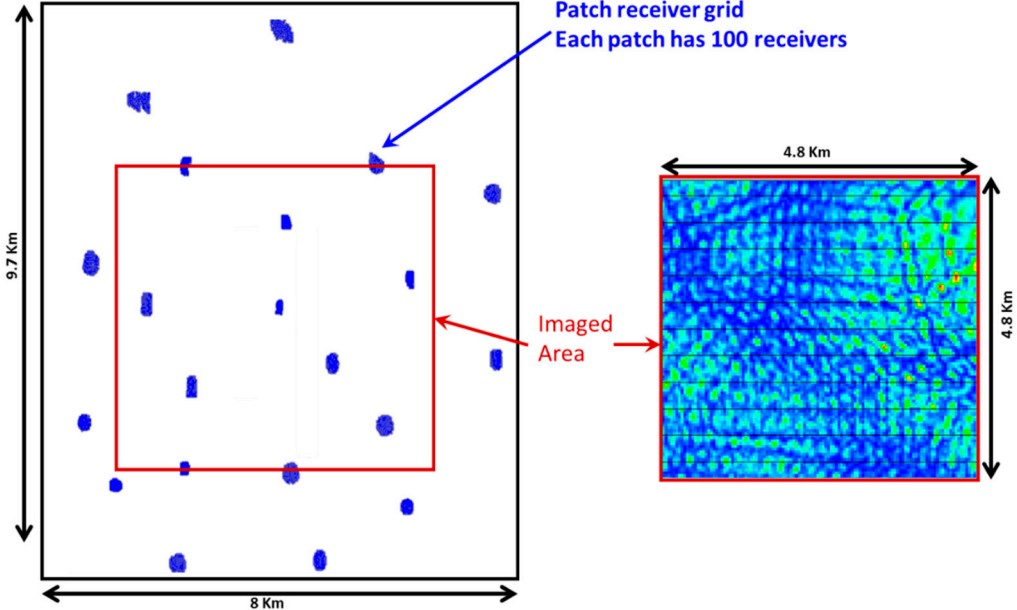

**Figure 6.** The patch receiver layout is designed such that there are 15 to 25 patches and within each patch there are 50 to 200 geophones. This layout allows for the suppression of surface wave noise within each patch and is focused on detecting and locating MEQs. For computing fracture seismic intensity volumes using one-way depth migration, this design is inferior. The geophone layout is shown on the left. A synthetic trace was computed for each geophone that would provide a uniform amplitude in the output fracture seismic intensity volume if a uniform grid was using the recording. When the synthetic is input to the one-way depth migration using geophone locations only at those for the patch geometry, the depth slice shown on the right is produced. The slice is for the area in the red box on the left. The patch geometry causes the short and long wavelength amplitude artifacts, and these will overprint any fracture patterns that may be imaged.

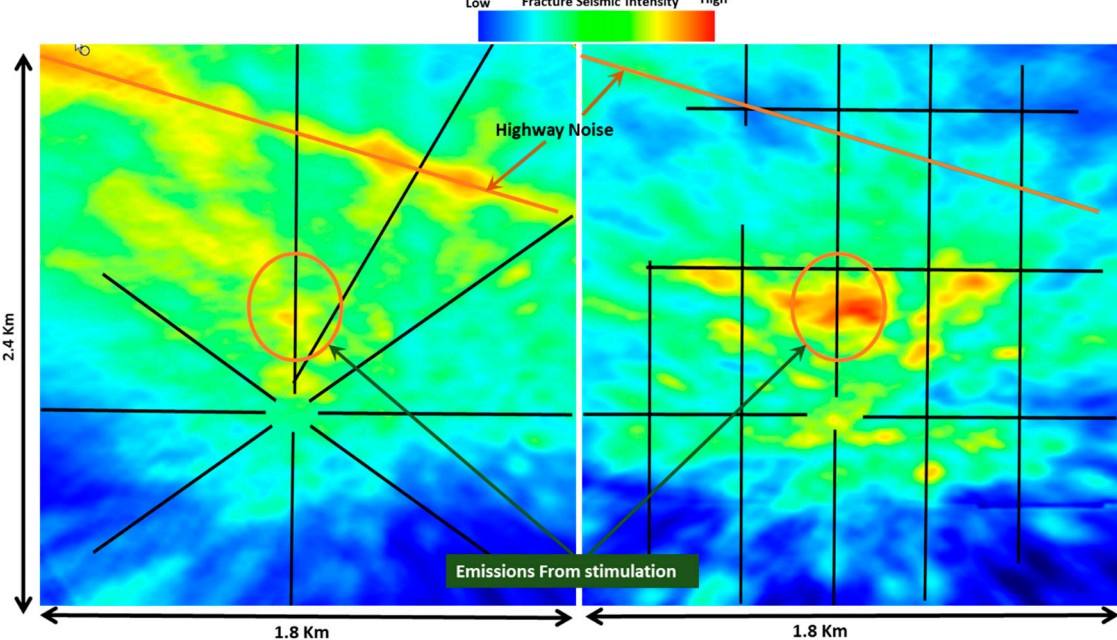

**Figure 7.** Highway noise in the star grid versus orthogonal grid. All data were recorded simultaneously so the signal content is the same for the star grid and the orthogonal grid. The star grid does not cancel the horizonal noise perpendicular to the star arm. The orthogonal grid suppresses the highway noise and the signal from the stimulation is enhanced.

## 2.4. Fracture Seismic: Signal Processing

The filtering and depth migration methods used for fracture seismic are based on typical reflection seismic signal processing algorithms, but are modified to deal with one-way travel times from the fracture seismic sources to the receivers. Success in using fracture seismic recordings for mapping fractures requires having high-quality non-resonant signal analysis and suppression (Sicking et al. 2016, 2017) [21,25].

The steps for processing fracture seismic can be broadly broken in to four parts: 1) Elimination of cultural and man-made seismic waveforms: 2) estimation of elevation and residual statics; 3) building the earth velocity model; and, 4) one-way travel time depth migration for a continuous signal source.

Cultural and man-made noise that is active for longer than minutes of time can be classified as stationary noise (Figure 8). This type of noise is common in industrial areas and transportation corridors and appears as noise background added to the consistent, slowly changing, harmonic character of fracture seismic signals. This long duration of noise can overwhelm fracture seismic signals, but the noise can be separated from the fracture seismic signals with cepstral filtering (Sicking, 2016) [21].

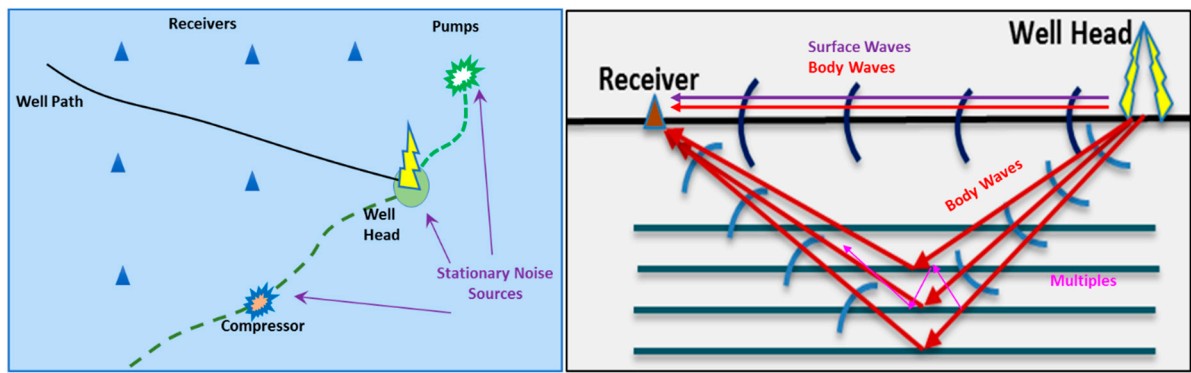

**Figure 8.** Continuous but erratic signals along the stationary source-receiver path can overwhelm fracture seismic signals. The left panel shows the locations for the receivers, the well head, and compressor noise sources. The right panel shows the ray paths from the well head to a single receiver for various types of seismic waves. The noise is generated at all times and the ray paths are fixed so the wave forms on the receiver trace are very repetitive. (Figure from Sicking et al., 2016) [21].

The cepstral filtering processes each fracture seismic trace independently by transforming the trace into the Cepstral domain, applying a bandpass in that domain, and inverse transforming back to time. The transform to the cepstral domain requires two forward Fourier transforms. The first Fourier transform is applied to the time data to compute the amplitude and phase as a function of frequency (a Fourier spectrum). The second forward Fourier transform uses only the amplitude versus frequency to compute the amplitude and phase as a function of quefrency (a Cepstrum). A low pass filter is applied in quefrency and the resulting amplitude and phase signal is inverse Fourier transformed to the frequency (Fourier spectral) domain. The amplitude versus frequency is combined with the original phase versus frequency before taking another inverse Fourier transform to compute the filtered trace in time.

The Fourier frequency spectrum for one trace of field data containing stationary noise is shown in the top panel of Figure 9. The erratic spectrum of the stationary noise is superimposed on the less variable, broader background of the more stable fracture seismic signal. The erratic part of the spectrum needs to be removed. Because of the large differences in their spectral character, the erratic stationary noise spectrum spreads to the full range of quefrencies while the fracture seismic signals are confined to the very lowest quefrencies. In fact, the low pass filter in quefrency needs only to keep the lowest 1% to 2% of the quefrency spectrum. The middle panel of Figure 9 shows the quefrency spectrum of the field data. A low pass filter is applied in quefrency that passes only the lowest 1% of

the quefrencies. When this filtered Cepstrum is reverse-Fourier transformed back to the frequency domain (bottom panel), the stationary noise is essentially eliminated.

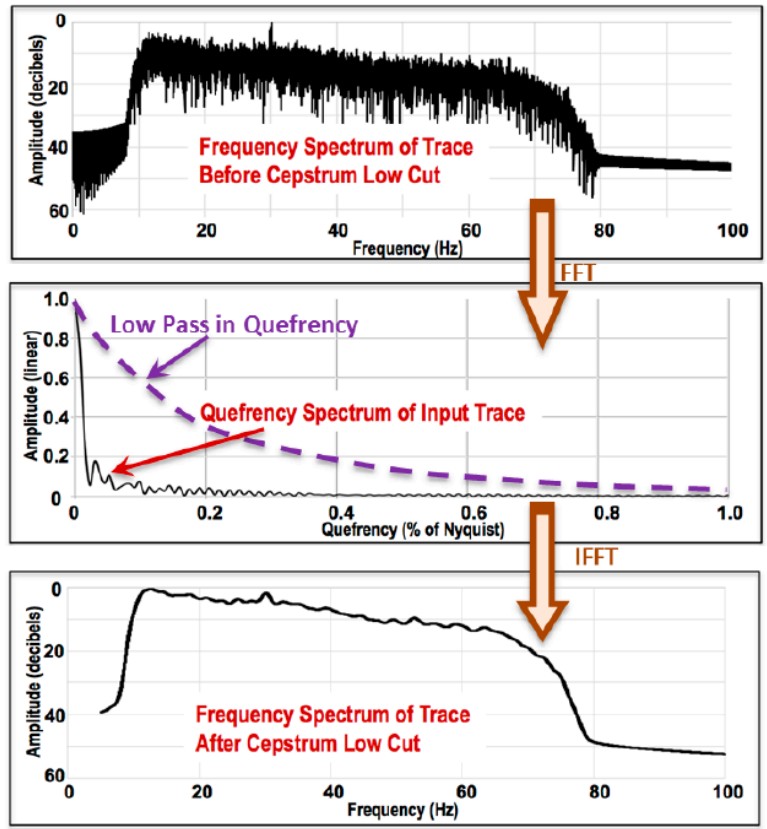

**Figure 9.** The cepstral filtering process to remove stationary noise. The stationary noise becomes spikes in the spectral domain (top panel). In the cepstral domain, these spikes are spread across all quefrencies (middle panel). The fracture seismic signals are in the lowest 2% of the quefrencies. After the application of the low pass filter in quefrency, the inverse transform to the spectral domain is shown in the bottom panel. (Figure from Sicking et al., 2016) [21].

The filter in the quefrency space has an amplitude of 1.0 at the smallest quefrency and an amplitude of 0.0 at the quefrency that is 3% of the Nyquist in quefrency space. More than 97% of the Quefrencies are thereby set to zero. Moreover, 95% of the trace energy is preserved in these lowest 2% of the quefrencies. Because the cepstral filter is a non-linear filter, its order of application is important: It does not commute with other linear signal processing filters. Experience shows that it should be run as the first filter in the processing flow.

The value of the cepstral filter in fracture seismic processing is illustrated in the top two panels of Figure 10. Here, the time window of noisy multichannel data includes the waveforms for a relatively large amplitude microearthquake (MEQ). The data are sorted by the azimuth direction with respect to the location of the MEQ. The traces have been shifted in time using one-way travel times from the voxel of the MEQ to each receiver on the surface. The waveforms for the MEQ should all arrive at the same time after the trace shifting. The figure shows the data before and after cepstral filtering. In the unprocessed data, the MEQ is not readily evident in the trace data. After cepstral filtering of each receiver trace, the MEQ signal emerges in the middle of the record section.

The suppression of other types of noise also aids in obtaining clear fracture images. These include filters that help identify and clean up transient amplitude bursts, electronic line noise, and traffic noise. The trace section in the bottom of Figure 10 shows the effects of further processing to reduce such interferences. The result clearly reveals the MEQ's signals, including its azimuthally dependent radiation pattern.

Another example of stationary noise suppression is shown in Figure 11. The top panel of this Figure contains a spectrogram of fracture seismic data. A 5-min section of constant 45 Hz noise is circled. Because this signal is continuous over time, one-way depth migration spreads it out in space in the fracture seismic intensity volume, as is shown in the left and right sides of the lower panel. The key for identifying this signal as stationary noise in the fracture seismic intensity volume is the alignment of the features it produces in the final processed volume. These features are linear and point back to the surface position of the noise source, the presence of which was later identified in surface maps and images. Thus, in addition to cepstral filtering, careful selection of time windows to avoid including stationary noise greatly aids in fracture seismic intensity mapping.

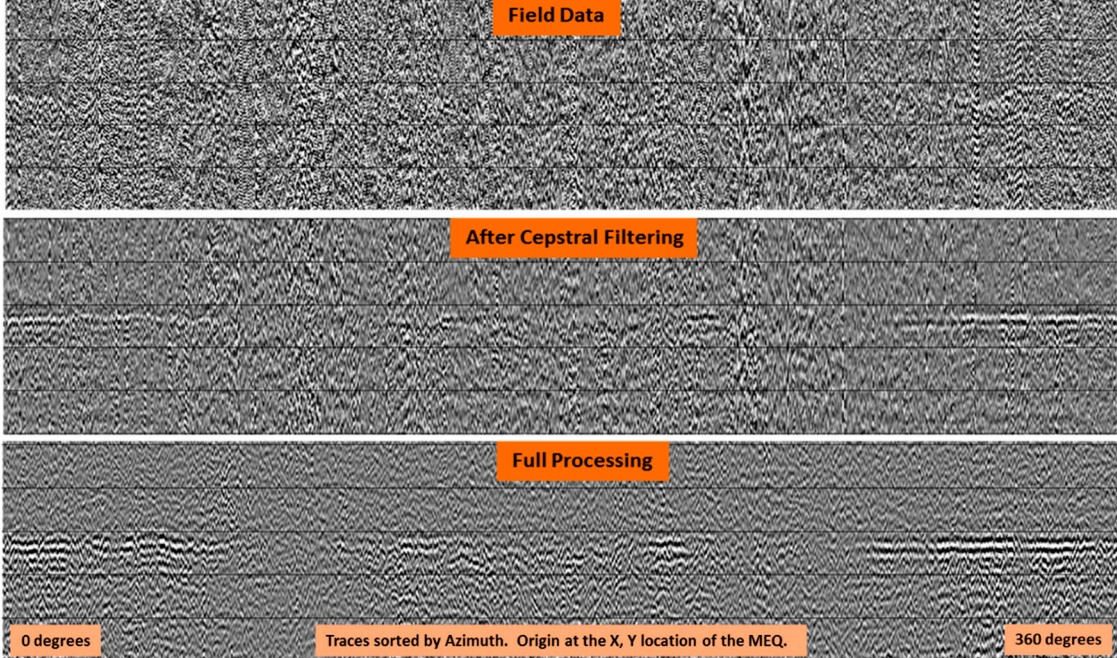

**Figure 10.** Cepstral filtering reveals the presence of a small microearthquake in these multichannel fracture seismic data. The top panel shows the traces as recorded in the field. The middle panel show that traces after cepstral filtering revealing the microearthquakes (MEQ). The bottom panel shows the trace data after all filtering has been applied. Figure from Sicking et al. (2016) [21].

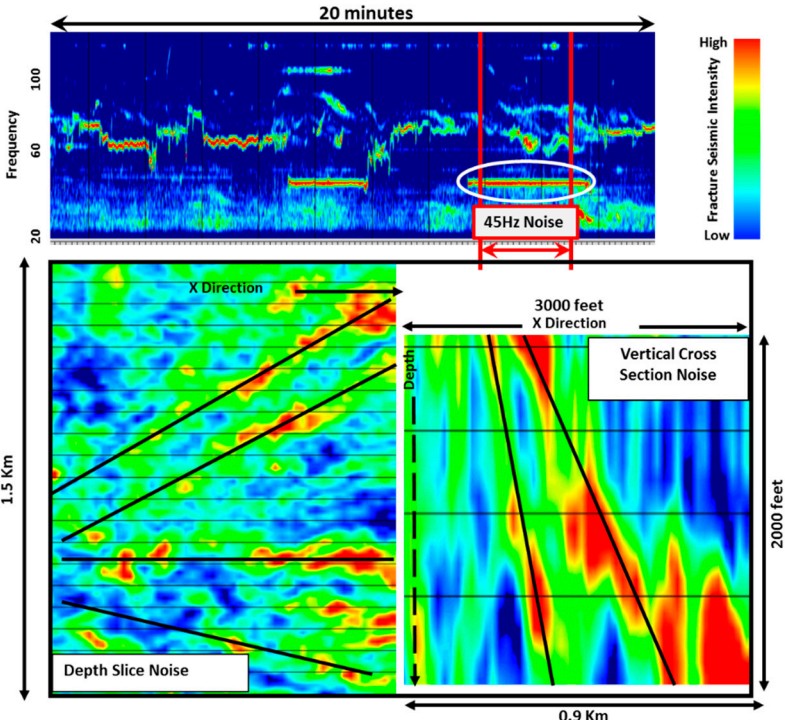

**Figure 11.** Surface noise in the traces appears in the spectrogram as narrow frequency band noise (top panel). The time window indicated by the red bars was used to compute a depth slice and a vertical slice of the fracture seismic intensity volume (bottom two panels). The narrow frequency band noise shown in the spectrogram causes the linear features in the fracture seismic intensity volume noted by the black lines. Tracking the black lines back to the intersections reveals the surface location of the noise source.

Elevation and residual statics are important because the one-way travel time depth migration assumes that the traces are shifted in time to approximate a flat elevation datum. Therefore, fracture seismic traces must be shifted to account for differences in the receiver elevations and for the near surface velocity variations. The elevation statics are computed by taking the difference between the surveyed elevation of the geophone and the chosen constant elevation datum, computing the travel time for the elevation difference using the near surface velocity. The computed travel time shifts are applied to the traces before depth migration. The optimum method for analyzing the elevation statics, the residual statics, and the correct velocity model is to record the waveforms from a perforation shot that is visible on all geophones. Using the initial velocity model, the one-way travel times from the X, Y, Z location of the perforation shot to each geophone are computed and applied to the traces after correction for the elevation differences. If the first break time of the perforation shot waveform is approximately at the same time for all of the traces, the velocity model and elevation statics are accurate. The top panel of Figure 12 shows that the arrival times on average are flat in time for the full offset range for this example. The remaining variations from the same arrival time are caused by near surface velocity differences between the individual receivers. The time shifts for each trace to get them to the same arrival times are the required residual statics. The bottom panel of Figure 12 shows the traces after correction for residual statics and shows that the waveforms arrive at the same time on all traces. By using the correct velocity and good quality statics, the depth migration can be computed with very high confidence that the intensity volumes will be good quality.

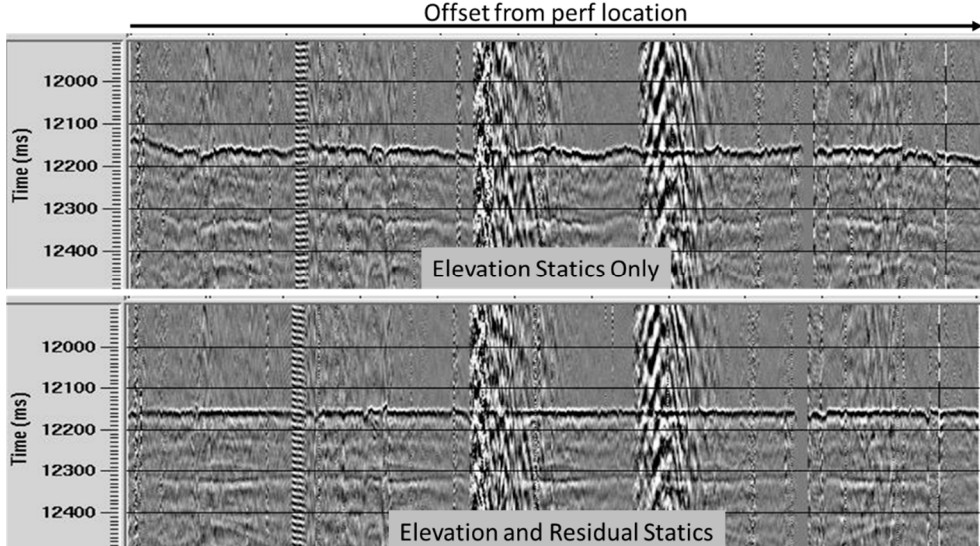

**Figure 12.** The top panel shows the traces sorted by offset from the X, Y location of the perf shot and with the time shifts applied using travel times from the perf location in depth to each receiver such that they should all be flattened at the same time. The traces are adjusted for elevation differences between the traces. The bottom panel shows the traces time shifted for the residual differences remaining on the traces in the top panel.

The velocity model must be accurate in order to obtain correct locations for the fractures in the volume. Often, a 1D velocity model is constructed from the sonic log recorded in a nearby well. This 1D velocity model is used to fill the entire 3D velocity volume such that the travel times are a function of offset only. This can work well for the small area around the well if the rock layer strata are flat lying and relatively homogeneous. For most areas of the Eagle Ford formation in Texas, there is a lateral velocity gradient such that for a constant depth the velocity decreases towards the Gulf of Mexico. When there is a lateral velocity gradient, using the same velocity for all voxels in the fracture seismic intensity volume results in a location error. Figure 13 shows an example of the velocity volume from the Eagle Ford that shows the gradient very well. When a 1D velocity model is used to focus and locate the perf shots, the gradient in the actual earth velocity causes errors in the locations of the perforation shots. The measured location errors for the perforation shots provide the information required to compute a set of statics that can be applied to the traces to force the location of the perforation shots to their known correct locations.

The location correction using the gradient will not work for areas where the velocity volume has 3D complexity. For areas with 3D complexities, the full 3D interval velocity must be used for all aspects of focusing and imaging in order to obtain useful results. A 3D complex velocity model is best derived using 3D reflection data and iterative pre-stack depth migration. Figure 14 shows an example of a complex 3D velocity model in a Colombia thrust zone. For fracture seismic, the one-way travel times in a complex velocity model must be computed using a full 3D ray tracing algorithm.

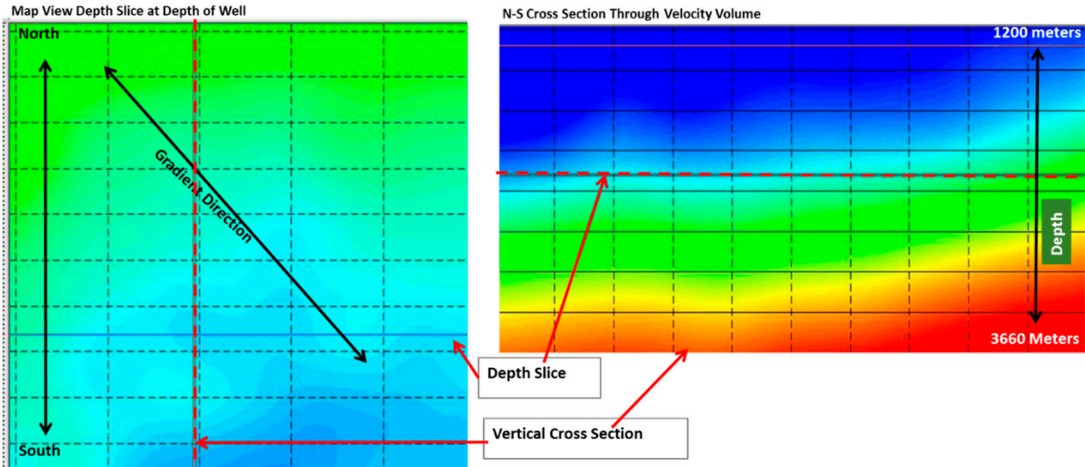

**Figure 13.** The velocity volume computed from the 3D reflection seismic shows a gradient. Using the 1D velocity model derived from the sonic log and this gradient, the perf shots can be positioned to match the known locations. Using this method to calibrate the location accuracy avoids the requirement to build a 3D interval velocity model.

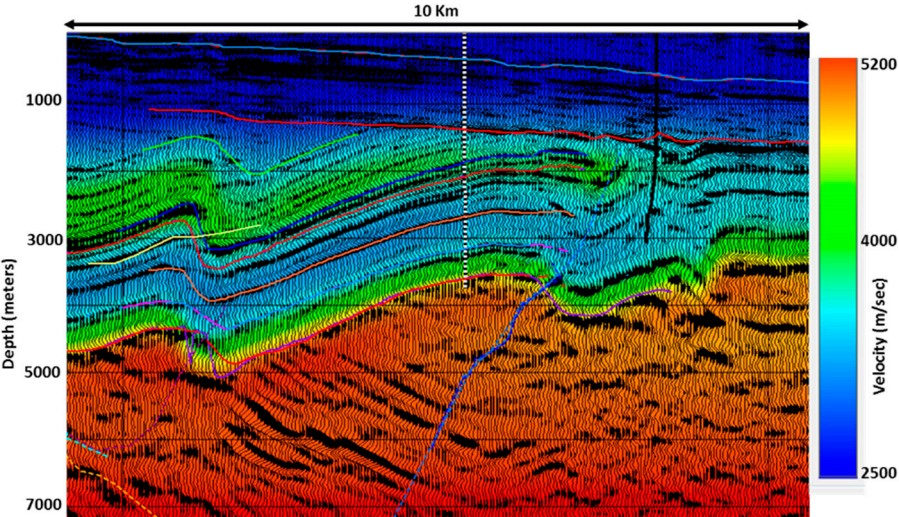

**Figure 14.** 3D complex velocity model in thrust zone computed using iterative depth migration. The fracture seismic intensity volumes computed with this velocity model will tie the reflection data and the fracture seismic intensity can be mapped onto the geologic structures.

*2.5. Identification of Resonance and Turbulent Flow in Fracture Seismic Trace Data*

Resonance and turbulent flow signals are identified in spectrograms computed from the trace data. The spectrograms are naturally noisy and extra care is taken to build up the signal during the computation of the spectrograms. For a surface location of interest, several traces very close to that location are selected for computing a single spectrogram. Spectrograms are computed for each selected trace and then the spectrograms are stacked. The first step in the computation of the spectrogram for a single trace is to compute the Fourier transform (FFT) for the first second of the trace and store the amplitude versus frequency at the first time sample in a two-dimensional array of frequency–time. The 1 s window is moved up in time by one sample, the FFT is computed, and the amplitude of the FFT is stored in the second sample along time. For trace data sampled every four-milliseconds, there are 15,000 one-second windows in 1 min of trace data. After the spectrogram is computed for every selected trace, all of the frequency–time arrays are stacked to obtain a single spectrogram for the location of interest.

Spectrogram analysis facilitates the efficient identification of time windows for use in computation of fracture seismic intensity volumes. Time periods with the strongest resonances are

selected from the spectrograms and used in the depth migration method to map their spatial locations (Sicking et al., 2016, 2017) [21,25].

A spectrogram computed for data recorded during the startup of the first stage stimulation for a well in the Eagle Ford is shown in Figure 15. The quiet time before the pumping is initiated shows some resonances that are episodic. They build in amplitude over time and they transition from dispersive to turbulent flow as the pumping continues and the pressure rises.

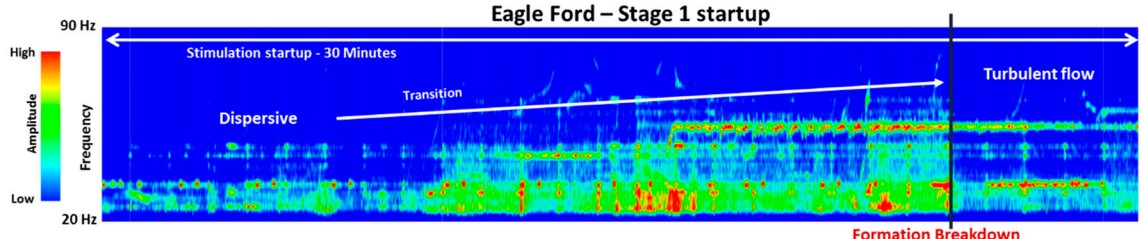

**Figure 15.** Resonance during stimulation. Eagle Ford during the startup of the stimulation for Stage 1. The resonances show a transition from dispersive to turbulent flow. There is a very pronounced change at the time when the formation breaks down.

Tary (2014 [3,4]) shows that fracture seismic resonances and turbulent flow are correlated with changes in pressure and slurry rate during stimulation. An example of this correlation is shown in Figure 16 where the spectrogram for 13 min of trace data recorded during stimulation is shown along with the pressure and slurry rates used for the same 13 min. The corresponding time windows between the treatment curves and the spectrogram are denoted by the vertical yellow lines. The resonance patterns in the spectrogram change at the same times that the pump curves show changes. This supports the interpretation that the resonances in the spectrograms are signal that is excited by the stimulation in the reservoir.

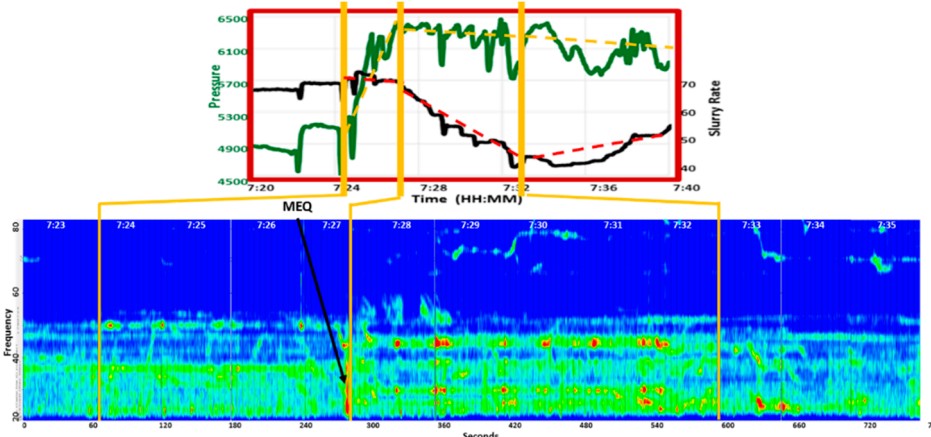

**Figure 16.** The spectrogram for 13 min of trace data recorded in the New Albany shale during stimulation correlates with the pressure and slurry rate curves. The pressure and slurry rate curves are shown in the top panel and the spectrogram is shown in the bottom panel. Four different time windows are denoted by the yellow lines and show that the changes in the treatment curves are correlated with changes in the spectrogram.

### 2.6. One-Way Travel Time Depth Migration

The fracture seismic intensity volumes are computed using Kirchhoff depth migration with one-way travel times. The one-way travel times are computed from each voxel at depth to every receiver on the surface. Kirchhoff migration is a two-step process that first applies a time shift to each trace equal to the travel time from the voxel to the surface geophone and then images across all of the time shifted traces.

An intensity volume is computed for each 200-millisecond time window of the trace data with a move up of 100 milliseconds between the intensity volumes. The intensity is computed for every voxel in the depth volume for every time window. This produces a new fracture seismic intensity depth volume at 100 millisecond steps.

Figure 17 shows a graphic of the for processing, focusing, and imaging used to compute the 3D depth fracture seismic intensity volume for each 100 milliseconds. The time interval for computation of intensity volumes can range from a few minutes to several hours. The imaging application must compute intensity volumes that can be coherently stacked over the entire time interval. Because of this stacking requirement, the fracture seismic intensity volumes are computed using semblance (defined in Figure 17) and the values in the intensity volumes are all positive. The phase of the waveforms in the trace data can change for each time window such that if the image computation method preserved the phase of the trace data, the fracture seismic intensity volumes from one depth volume to the next would not stack coherently.

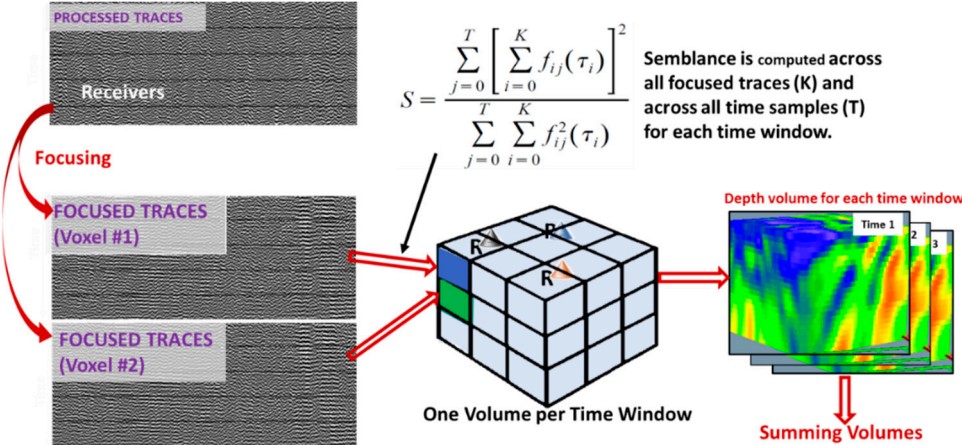

$$S = \frac{\sum_{j=0}^{T}\left[\sum_{i=0}^{K} f_{ij}(\tau_i)\right]^2}{\sum_{j=0}^{T}\sum_{i=0}^{K} f_{ij}^2(\tau_i)}$$

Semblance is computed across all focused traces (K) and across all time samples (T) for each time window.

**Figure 17.** Workflow for one-way travel time depth migration. After trace processing and velocity model building and calibration, the traces are depth migrated for each time window and each depth voxel for the time interval that will be summed; $f_{ij}$ is the trace amplitude at trace I and time j.

The differences between the method described by Kochnev (2007 [9]) and the one here are that they compute the coherency only for a subset of the depth volume and for previously identified time intervals. The streaming depth migration described here streams all of the trace data for the entire time interval of interest through the processing workflow and computes tens of thousands of depth volumes

The post migration processing identifies the time windows in which coherent noise contaminates the fracture seismic intensity volumes and deletes them. The time windows in which large amplitude MEQ occur are detected and deleted. The final fracture seismic intensity volume is computed by summing all of the volumes that are not deleted.

*2.7. 3D Fracture Extraction Methods*

Fractures are extracted from the fracture seismic intensity volume by first picking all of the local maxima in the volume. Two methods that are currently employed are picking the local maxima and computing the value of the maximum negative curvature. The voxels that have local maxima or maximum curvature are connected to each other to form complex 3D surfaces that show the connectivity of the permeable fractures throughout the volume. Copeland et al. (2015) [35] describes the curvature method for tracking the maximum curvature in the intensity volume.

Petrophysical and engineering measurements support this interpretation for data recorded when there is no industrial activity (quiet times) and for data recorded during stimulation (Sicking et al., 2016, 2017) [21,25], (Geiser et al., 2012 [26]), (Lacazette et al., 2013 [27]).

## 2.8. Location Accuracy – Correlation with Distributed Acoustic Sensing (DAS)

The fracture systems are very complex 3D surfaces and there is always the question concerning the location accuracy of these surfaces. The location accuracy of 3D reflection seismic imaging measures the offset in three dimensions for reflections and faults from the locations determined from drilling. Fracture location accuracy can be measured using some of these same methods.

The location accuracy of images from one-way travel time depth migration is on the same order of magnitude as that obtained by reflection seismic imaging because it employs the same band pass in frequency and the same velocity model. Fracture seismic fracture locations have better location accuracy because there is integration over long time periods and the accumulation of signal over the integration time improves the location accuracy. Reflection seismic imaging does not have this advantage.

Figure 18 shows a synthetic study that demonstrates the improvements obtained from integration over time. A fracture was modeled in a 3D velocity volume and the signals emitted from the fracture are very small. However, the emissions from the fracture continue episodically for 15 min. The recording system has 2000 receivers on the surface and the fracture is located at 5000 ft. depth. The noise in the trace data has sufficient amplitude that the signal in not visible in the traces.

The images in Figure 18 shows a depth slice of the intensity volume through the fracture. As the integration time increases, the Signal-to-Noise ratio (S/N) increases and the resolution of the fracture improves in that the peak signal to background noise increases and the measured width of the fracture narrows. With sufficient integration time, spatial location of the fracture reaches an accuracy of 8 to 15 m.

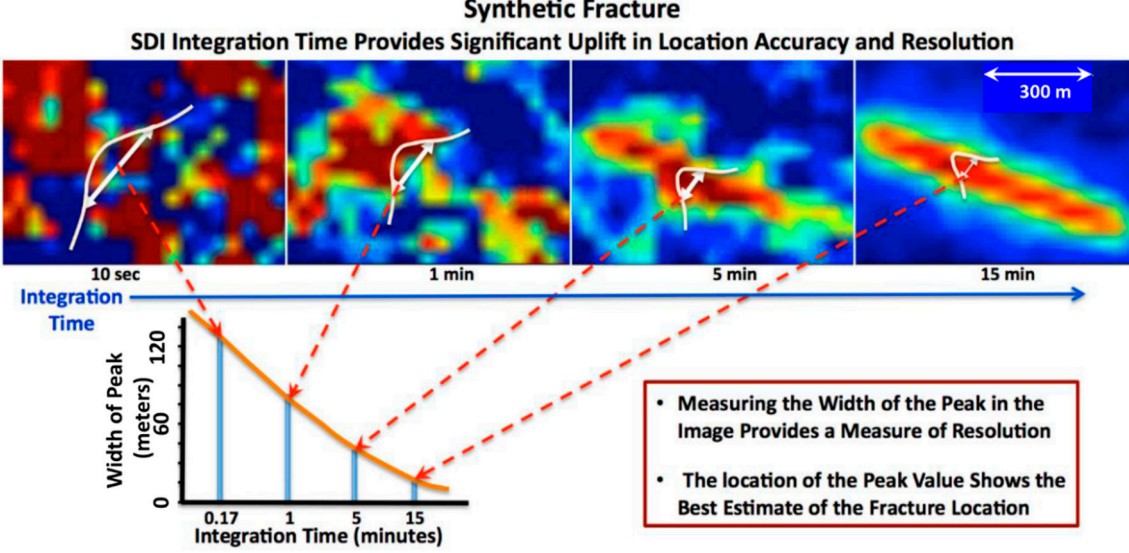

**Figure 18.** Integration of fracture seismic volumes increases the location accuracy, the S/N, and the resolution. For 10 s of integration, the fault is not visible. After 1 min of integration, the fault begins to be recognized. After 5 min, the fault is well defined and after 15 min, it is well resolved.

Figure 19 shows the comparison of a fracture image from fracture seismic data integrated over the entire stage with the fiber optic cable acoustic recording for the same stage. The acoustic signal from the fiber optic log shows that most of the frack fluid came from the perf location nearest to the well head. The fracture seismic fracture image crossed the well within 3 to 8 m of that perf location. This is a direct comparison of two independent measurements and shows that the location accuracy of fracture seismic is very good.

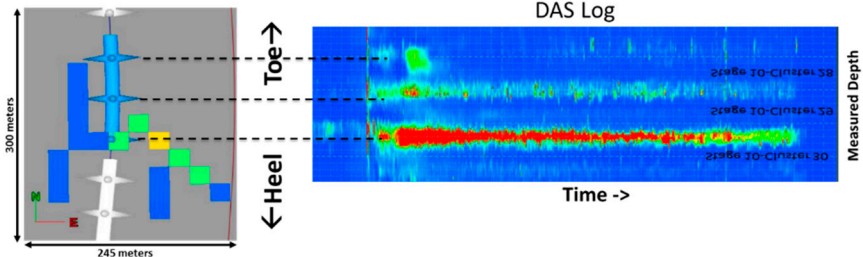

**Figure 19.** Comparison of the fracture image from fracture seismic traces and the distributed acoustic sensing (DAS) log from a fiber system recorded during the stimulation. The DAS plots and the fracture image show the same result, which supports the interpretation that the location accuracy of fracture imaging is on the order of 8 m.

### 2.9. Fracture Seismic Images and Hypocenters/MEQ

Fracture seismic was recorded using a buried grid in the Eagle Ford during the stimulation of four wells. The hypocenters were detected and located for the stimulation times for all stages of all wells and are shown in the right panel of Figure 20. The fracture seismic method was used to extract fractures for the same treatment times as was used for the hypocenter detections and they are shown in the left panel of Figure 20. In most projects where this comparison is made, it is rare that there is a direct correlation between detected hypocenters and the fracture seismic maps.

The waveforms that are emitted from the fracture tips for the hypocenters and MEQ are sourced by different mechanisms than the signals used to compute the fracture seismic intensities. The interpretation of this phenomenon is that the fracture seismic energy from fluid-filled fractures are emitted along the length of the fractures while the hypocenters occur at the tips of the fracture. Thus, the larger MEQ are not collocated with the fracture seismic image and the smaller opening mode hypocenters are at the tips of the fractures.

These differences in source mechanisms and signals explain why the fracture models computed using hypocenter locations are almost always different from the complex 3D fracture models derived using fracture seismic methods.

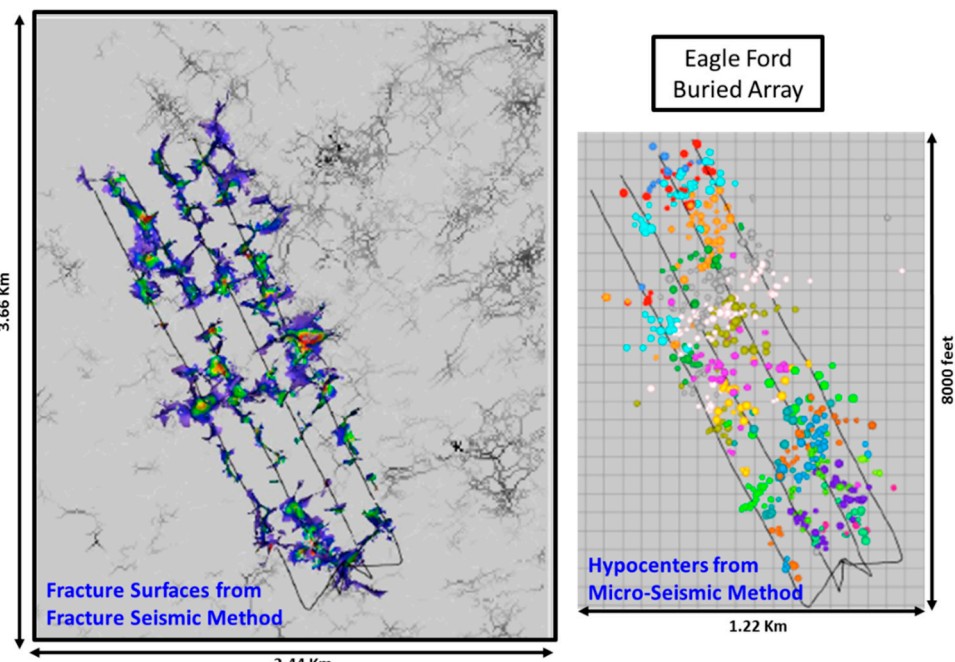

**Figure 20.** Comparison of fracture images from fracture seismic to the MEQ locations in the same data. The fracture seismic data were recorded using a buried array in the Eagle Ford. There is only a partial correlation between the fracture surfaces and the MEQ.

## 3. Case Studies

A large database of case histories demonstrates the capacity of fracture seismic methods to directly map fluid-filled fractures and their role in subsurface connectivity. The examples come from different basins and different geologic settings They also come from data recorded before, during, and after various kinds of industrial activities, and from both greenfield as well as brownfield sites.

Fracture seismic methods have been used to map the fracture connectivity during the stimulation of approximately 100 horizontal wells with almost 2000 stages. Fracture seismic observations before, during, and after these stimulations show that the fracture systems that produce the most fluids are the same fractures that are mapped before wells are drilled. We will also show how fracture seismic can track the fluid producing volume over the life of the well.

Figure 21 shows a subsurface rectilinear volume that contains a well for which the fracture seismic method was applied before, during, and after the stimulation of the well. The lower panel shows a depth slice of the intensity volume computed before the stimulation. The back panel show a vertical slide of the intensity volume computed during the stimulation. The 3D volumetric view in the center shows the producing volume for the well after it was being produced. The actual volume of rock that is producing fluids is quite different from the stimulated rock volume computed during the treatment.

In addition to local hydraulic fracture stimulation projects, 15 larger-scale fracture seismic mapping studies have been completed. These include projects when the fracture seismic is recorded before drilling, stimulation, production, subsurface flooding, or other related industrial activities are underway. The correlation between the fracture systems computed for quiet times and the fracture systems computed during production indicates that the fracture systems are excited by pressure changes caused by natural earth processes such as tectonics or earth tides as well as by the stimulation treatments.

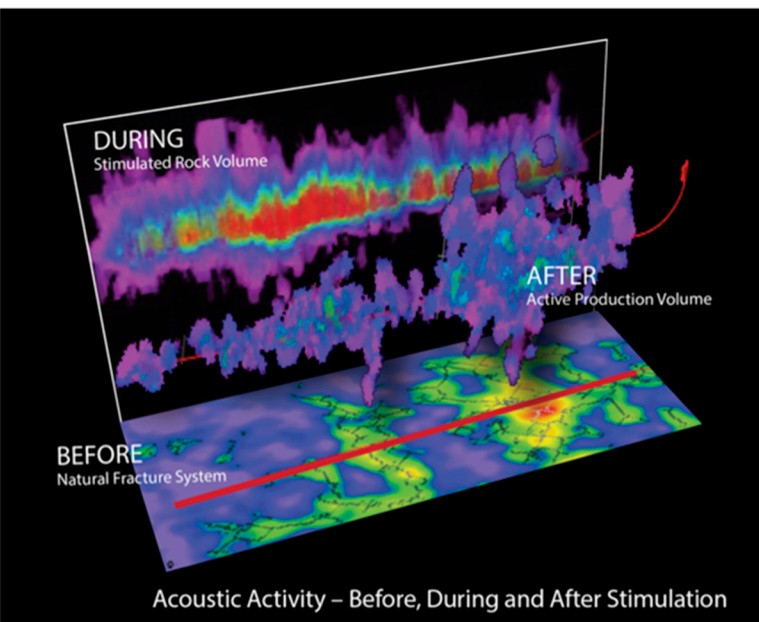

**Figure 21.** Fracture seismic imaging of the subsurface is applied before, during, and after hydraulic stimulation.

### 3.1. Intensity Burst during Stimulation—Texas

This example shows fracture seismic signals that are distributed along the length of a single fracture. These observations were recorded during the hydraulic stimulation of one stage in the Eagle Ford shale of Texas. It consists of a single burst of high-energy resonances that lasted for 7 s (Figure 22). The burst was recorded with a surface grid laid out with orthogonal cables that had 2100 receivers over an area of 65 square km.

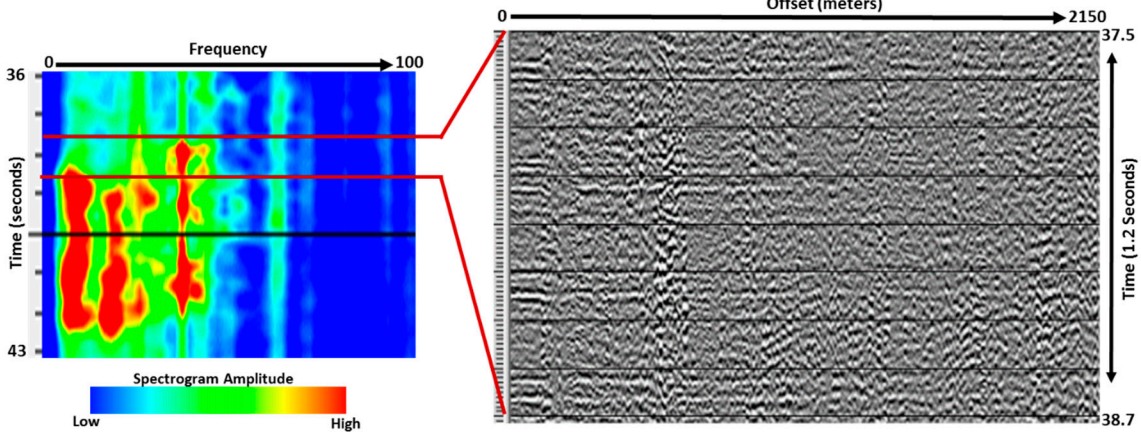

**Figure 22.** Intensity bursts during stimulation. A burst of higher amplitude waveforms in the trace data continued for 7 s. The spectrogram of the 7 s is shown on the left. The amplitude of the signals was sufficiently high that the waveforms are clearly visible in the individual traces as shown in the plot on the right.

The spectrogram in the left panel of Figure 22 for these 7 s shows the narrow band energy that we initially identify as resonance from turbulent flow into the fractures surrounding this stage. The seismic record sections on the right show that the amplitudes of this burst can be seen on the individual receivers with amplitude that is well above the background noise. The waveforms do not change phase with azimuth or offset, which indicates that this is not a point hypocenter or MEQ.

Fracture seismic depth migration was used to map the 7 s burst to the location of its source. The fracture seismic intensity volume shows that all this energy came from a very small area (see Figure 23). The resonance shown in the spectrogram and the vertical extent of the source location supports the interpretation that it is a resonance initiated either by flow into the fracture or by resonance along the entire fracture. The overlay of the fracture seismic intensity on the associated 3D seismic reflection section shows that it is in a small synclinal structure and is oriented in the vertical direction.

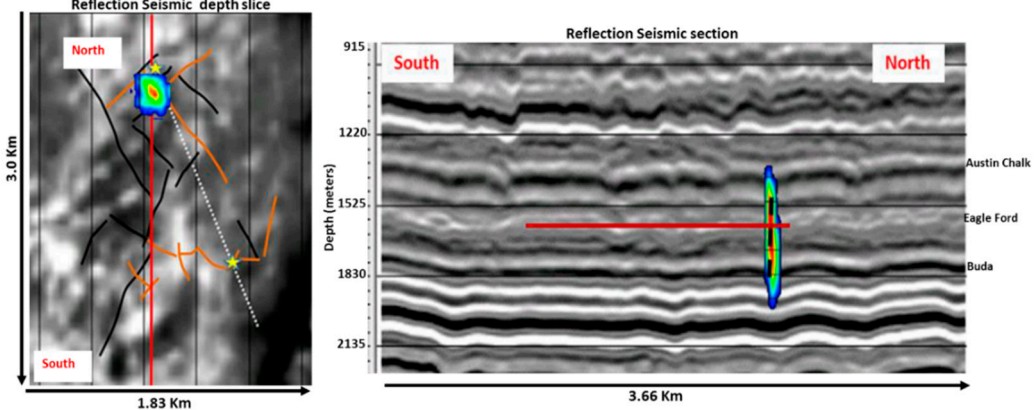

**Figure 23.** Computing an integrated fracture seismic intensity volume over the 7 s of data in Figure 22 and plotting on the reflection seismic shows that the energy came from a fracture in a small syncline that extends in depth from the Buda to the Austin Chalk and is focused into a very small area just to the West of the well near the stage being stimulated.

## 3.2. Thrust Fault Activation during Stimulation—China

This example shows that faults can have both permeable zones and zones that do not transmit pressure or fluids. A horizontal well drilled along a reservoir layer in a thrust zone is shown in Figure 24. This well was parallel to and 300 m shallower than the thrust fault mapped on reflection seismic. Given this vertical separation, it was thought that hydraulic stimulation of the well would not affect

this structure. In order to evaluate the stimulated resource volume of this treatment, fracture seismic data were recorded using 1600 receivers in a surface grid.

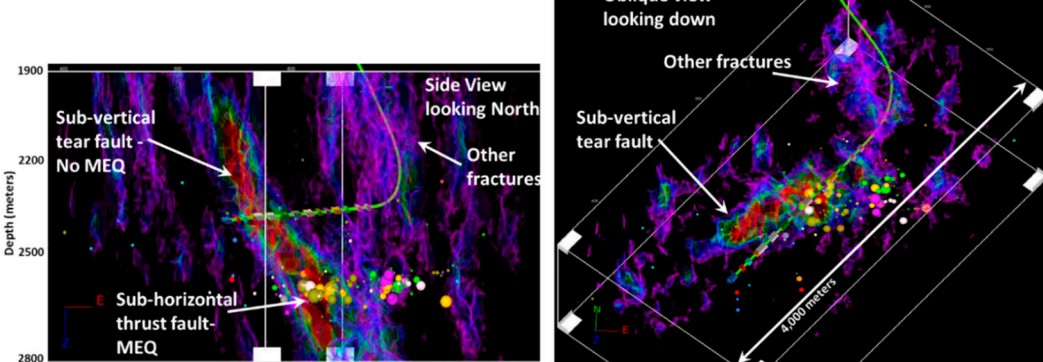

**Figure 24.** MEQ in the trust fault that is 300 m below the well were activated during only four middle stages of the well treatment. The pressure from the stimulation propagated along the tear fault down to the thrust and caused activation in the thrust zone. The pressure from the stimulation activated resonances in the tear fault so that it was mapped in the fracture seismic intensity volume. The side view of the fracture image volume (left panel) shows the tear fault, the well, and the MEQ in the trust fault. The oblique view (right panel) show that the tear fault is very close to the well (50 m).

The recording for each stage shows that there is a zone in the nearby tear fault that is permeable for only part of the length of the well. The permeable zone is active for only four of the stages during the treatment. For stimulation stages near the toe and near the heel of the well, the trust fault was seismically inactive. During the tear-fault-related four stages, the stimulation pressure was transmitted down to the thrust fault and caused many larger microearthquakes. None of the other treatment stages cause microearthquakes in the thrust. Evidently, the fracture seismic imaged permeable zone transmitted the pressure from the pumping to the thrust and a large number of MEQ are stimulated during the four stages.

The side view in Figure 24 shows the width and height of the tear fault and that is permeable. The tear fault is approximately 50 m from the well path. The fracture image volume in Figure 24 shows the tear fault width and height. The width mapped is the same width and location as the location of the four stages. The height of the tear fault goes vertically above the well and below the trust fault depth.

### 3.3. Large Single Grid in Thrust Zone, No Stimulation—Colombia

This example shows how fracture imaging using fracture seismic is integrated with 3D reflection seismic in a new area to select drilling location (Sicking et.al., 2017) [25]. This large area in Colombia had not been actively explored for over 60 years. The reservoir in this area is the Rosa Blanca formation that produces gas in areas where it is naturally fractured. During the decade of the 1950s, this area had been drilled with little success. There were many dry holes but one well struck a highly fractured zone and blew out. Subsequently, the area was abandoned.

In 2012, before initiation of new drilling, a modern 3D reflection seismic survey was collected and, as an independent acquisition, a 4650-channel fracture seismic dataset was recorded that covered 50 square km. The fracture seismic area was in the middle of the area covered by the reflection seismic survey. Before any wells were drilled, the reflection seismic data was processed and migrated using pre-stack depth migration and a complex detailed 3D interval velocity model was computed (Figure 14).

The fracture seismic was processed using the fracture seismic method and a fracture seismic intensity volume was computed that included data from 15 h of recordings. The depth tie between the active and fracture seismic data was ensured by the use of the same interval velocity model for both depth migrations. The intensity volume was integrated with the reflection seismic depth

migration volume in order to find the structural positions of the most active fractures in the Rosa Blanca.

The fracture seismic traces were analyzed for resonance using spectrograms (Figure 1). The high fracture seismic intensity and resonance observed at this site is likely due to the high stress state caused by the compressional forces in this region of the northern Andes. The spectrograms help to identify time periods for computing the fracture seismic volumes. While the behavior of the resonances is not yet well understood, experience dictates that very active resonance time periods are the best times to use in fracture seismic depth migration to compute the intensity volume.

Figure 25a shows a map view of the horizon slice extracted along the target horizon from the fracture seismic intensity volume. It is overlaid with the structural contours interpreted from the reflection seismic volume for the same horizon. The horizon map shows that the highest intensities are in the hanging wall of the thrust fault. The black symbols are the locations of dry wells drilled in the 1950s. They are all in the low fracture seismic areas of the Rosa Blanca. The small red circle shows the location of the well that blew out in the 1950s. This well is at the top of the structure and in a higher fracture seismic area, which indicates fracturing on the Rosa Blanca. A new well shown by the large black circle was drilled into the hanging wall of the thrust fault and in an area of high fracture seismic.

Figure 25b shows the vertical cross section of the reflection seismic and the fracture seismic intensity volume through the new well. The reflection seismic data are in black and the overlay of the intensity in color. The intensities show that the hanging wall of the thrust is the most active and that the structure in the fracture seismic volume follows the structure in the 3D reflection data.

Integration of fracture seismic, together with interpretation of reflection seismic, can be used to indicate potential areas for drilling. Based on these data, a test well was drilled, and a heavily fractured reservoir was found in the zone of high intensity. From when the well was put on production and until April of 2019, this well has produced 1.7 BCF gas. Based on the results of Well X, we conclude that the high fracture seismic intensity along the flank of the thrusts does indicate areas of active fractures. By active, we mean seismically active with the implication, based on experience, that the fractures are also permeable.

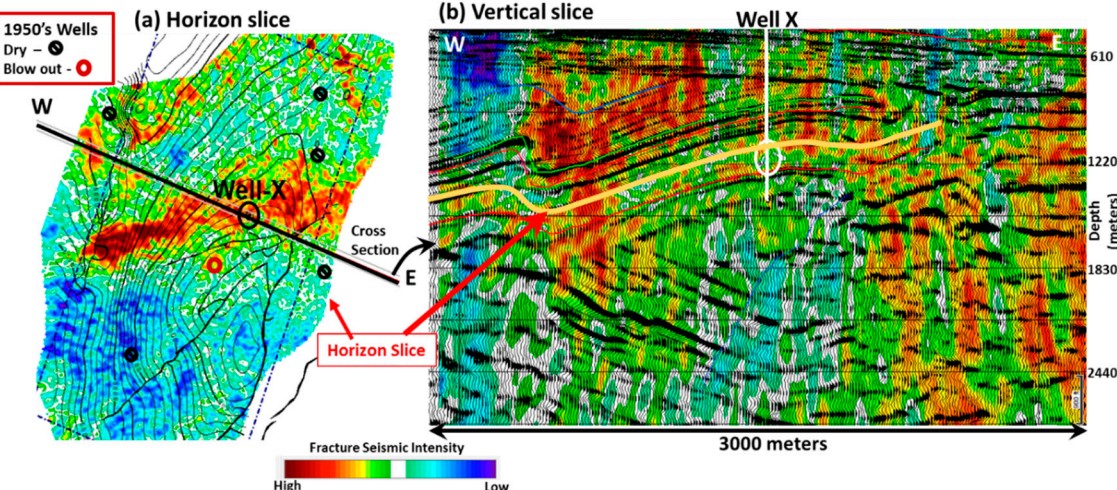

**Figure 25.** Identifying zones of fracturing in reservoir before drilling. Panel (**a**) shows the fracture seismic intensity extracted from the volume along the reservoir horizon with the overlay of the structural contours interpreted from the 3D reflection seismic. The traces of the thrust faults are shown. The highest fracture seismic are parallel to and out in front of the thrust fault. Panel (**b**) shows a vertical cross section through well X with the fracture seismic overlaid on the reflection seismic. The yellow line shows the horizon that was extracted to get the slice shown in (a). The fracture seismic intensity is higher down dip from where well X was drilled. Dry holes (small black circles) drilled before this analysis are in zones of low fracture seismic intensity. An old well that blew out (shown by the small red circle) and Well X are in zones of high fracture seismic.

*3.4. Roll Along Fracture Seismic during 3D Reflection Seismic Acquisition—Texas*

This example shows how fracture seismic can aide in selecting drilling locations. A 3D reflection seismic survey was collected in the Permian basin of west Texas using a nodal recording system. There were approximately 6500 simultaneously active nodes in the receiver array.

The fracture seismic data were collected concurrently with the reflection seismic by programming the nodes to record at night for 2 h when the crew was not shooting. This recording schedule meant that a very large area was covered by the fracture seismic for each day of recording. Each day of the fracture seismic recordings were processed as an independent data set. Multiple days of recording were selected such that there was substantial overlap in the independent intensity volumes. Nine hours of recordings over the area of the proposed horizontal well were selected for fracture seismic intensity computation.

Map and cross section slices of the fracture seismic intensity volume are shown in Figure 26. The well shown in Figure 26 was not drilled until two years after the seismic acquisition. The depth slice shows that the volume has areas of high intensity and areas of low activity. The proposed well path is predominantly in a zone of low activity with some higher activity at its toe and heel.

Shown in both the depth slice and the vertical slice along the well path is the log for the volume of acid uptake during the stimulation. This log shows that the acid uptake is highest in the zones that had higher fracture seismic intensity and was lowest in the zones of low fracture seismic intensity. They are consistent with the interpretation that the zones of higher fracture seismic intensity have higher fracture density and connectivity. This well was not economic. If the fracture seismic intensity volume had been used to plan the well, it could have been relocated to a zone of high-density natural fractures.

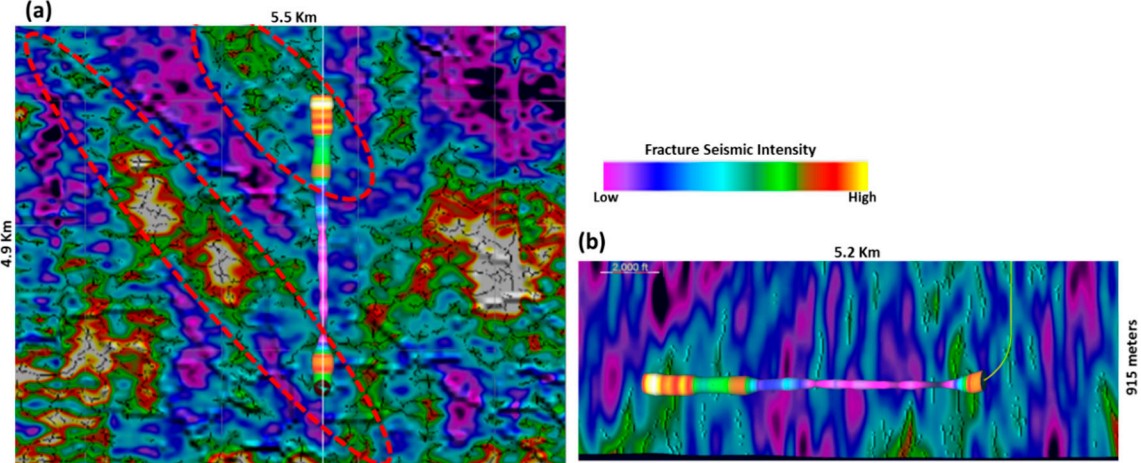

**Figure 26.** Slices of the fracture seismic intensity volume that was computed from data recorded during a 3D reflections seismic acquisition before the well was drilled. The depth slice (**a**) of the fracture seismic intensity volume is shown at the depth of the proposed well. The vertical slice (**b**) is along the path of the proposed well. The logs of the acid uptake show that the uptake was highest in the zones of highest fracture seismic intensity and lowest in the zones of low fracture seismic intensity. This result supports the interpretation that the fracture seismic intensity shows the zones of natural fractures.

*3.5. Roll along Fracture Seismic during 3D Reflection Shoot—Wyoming*

This example shows how fracture seismic is used for areal evaluation before development. A 3D reflection seismic survey was recorded in Wyoming that covered a very large area. A smaller area was selected for the fracture seismic recording. The entire receiver array of 6000 geophones was used for recording several hours on seven different days during the active seismic acquisition. The fracture seismic was recorded such that there was a 50% overlap in the fracture seismic intensity volume

between recordings. Each recording time was processed as an independent project and the seven fracture seismic intensity volumes were merged later.

The depth slice at the reservoir depth from the fracture seismic intensity volumes for all seven recordings is shown in Figure 27. The slice shows a fault zone across the Northern part of the volume that is also mapped in the 3D reflection seismic. There are two large areas of high fracture seismic intensity in the SW and NE corners of the survey area. These areas are separated by a NW to SE trend of lower fracture seismic activity.

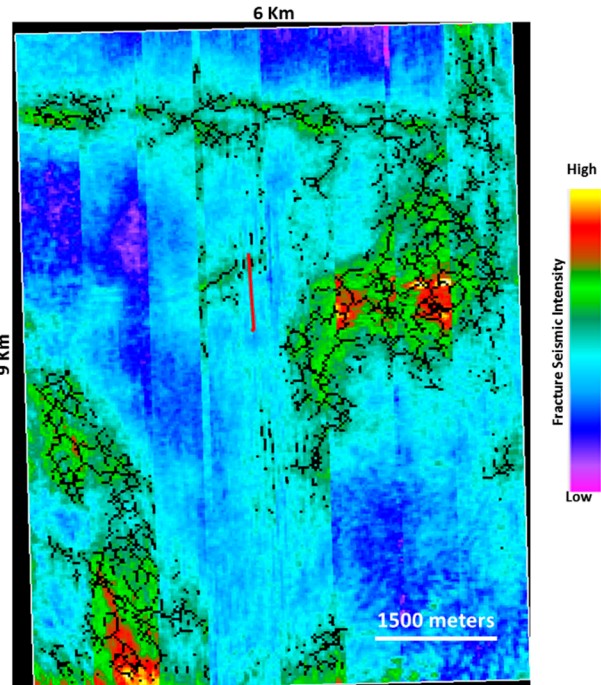

**Figure 27.** Depth slice of the fracture seismic intensity volume that was computed from fracture seismic recorded during a 3D reflections seismic acquisition before the well was drilled. The seams in the final volume are the result of merging the seven independent volumes. The linear feature that runs East to West in the North of the Volume is also seen in the 3D reflection volume.

The seams in the final merged volume are readily apparent in the depth slice shown in Figure 27. The fracture seismic intensity volumes are from data that were recorded a few days apart. The fracture seismic intensity during one day of recording is not the same for the other days. Differences in the fracture resonances can account for much of the differences. Processing may also account for small differences in the amplitude from one day to the next. It should be noted that there was a 50% overlap in the intensity volumes from one day of recording to the next such that every voxel in the volume has contributions from two days. Even with the presence of the seams, the fracture seismic intensity volume reveals important information on the natural fracture zones of this prospect and the optimal locations for drilling and stimulation.

### 3.6. Roll Fractures and Well Treatments—Illinois

This example shows how pre-existing fracture systems impact stimulation. Pre-stimulation fracture seismic was recorded over a well site in the New Albany shale after the well was drilled, but a few weeks before the well was stimulated. These data were used in the planning for the stimulation. Fracture seismic was also recorded during the stimulation. This allowed for the comparison of the fracture seismic intensity volume before, and after stimulation.

Spectrograms were computed from the fracture seismic data recorded pre-stimulation and from the fracture seismic data recorded during stimulation. Samples for both are shown in Figure 28. The top panel shows the spectrogram for a time window pre-stimulation and reveals a narrow band resonance in the 50 Hz to 60 Hz range, along with a broad distribution of signals at lower frequencies.

For comparison, the spectrogram for a time window during the stimulation is shown in the lower panel of Figure 28. The resonances are mostly in the lower frequency bands but have substantial changes in amplitude, character, and frequency band. These changes correlate with the stimulation pressure and fluid rate pump curves (Figure 16).

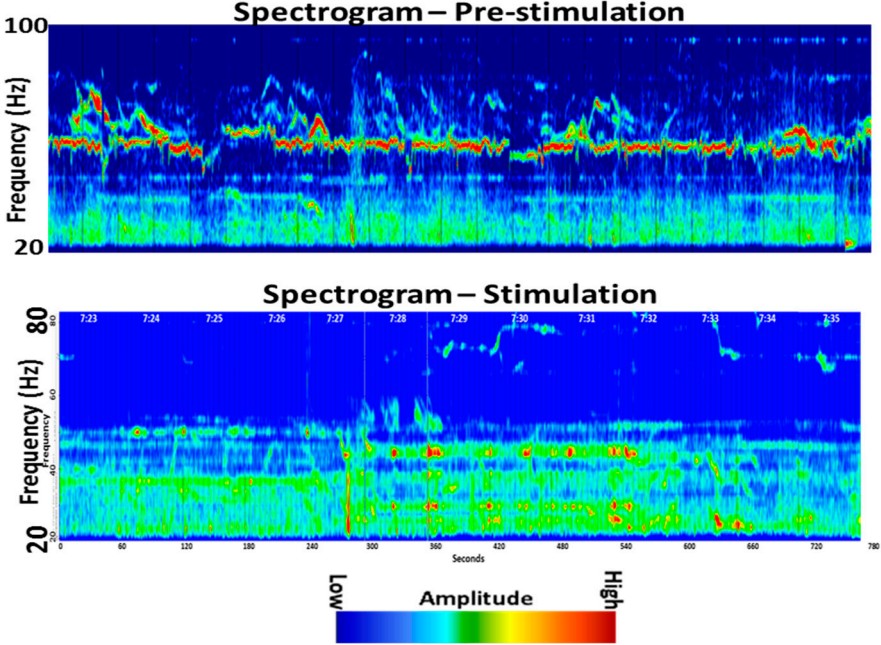

**Figure 28.** Spectrograms computed from pre-stimulation fracture seismic data (top) and data recorded during stimulation (bottom).

The fracture seismic intensity volume computed for this pre-stimulation time period shows that the fracture seismic activity is very high at the toe of the well and has significant zones of low activity in the middle of the well path (Figure 29, left panel). The fracture seismic volume from the pre-stimulation time period was used to plan the stimulation. During the first three stages of the stimulation, problems were encountered in getting the fluid to flow into the formation. The pre-treatment data were used to analyze the stress field to determine that the pressure used in the pumping should be reduced to solve this problem.

Comparing the pre-stimulation fracture seismic (left) to the fracture volume (middle) that was computed during the stimulation shows that the fractures that are computed from the data recorded during the stimulation follow the intensity patterns in the fracture seismic volume computed pre-stimulation. The overlay of the fractures computed during stimulation on the fracture seismic intensity computed pre-stimulation show that the pre-existing fracture system impacts the performance of the stimulation with the highest intensity zone during the pre-frack time having the highest density of activated fractures during the stimulation.

Using the fracture seismic intensity volume computed from the pre-stimulation and the volume computed during stimulation we forecast the connectivity pathways (Figure 29, right panel) in the reservoir that will produce the most fluid flow during production. These were computed by first thresholding the amplitude in each of the intensity volumes for each stage of the stimulation. In each voxel for each stage, the amplitudes that were below this threshold were reset to zero. For each voxel, the number of volumes that were above the threshold for that voxel were counted and the count for that voxel was stored in the repeated activity volume.

For example, with 16 stages, each voxel has the possibility of containing a number between 0 and 16. The connectivity pathways are then seen as the highest number of threshold crossings. This attribute volume is used to model the pressure and fluid transmission through the reservoir. The connectivity pathways for this example show that zones of best connectivity in the reservoir are the most active in the pre-stimulation fracture seismic volume.

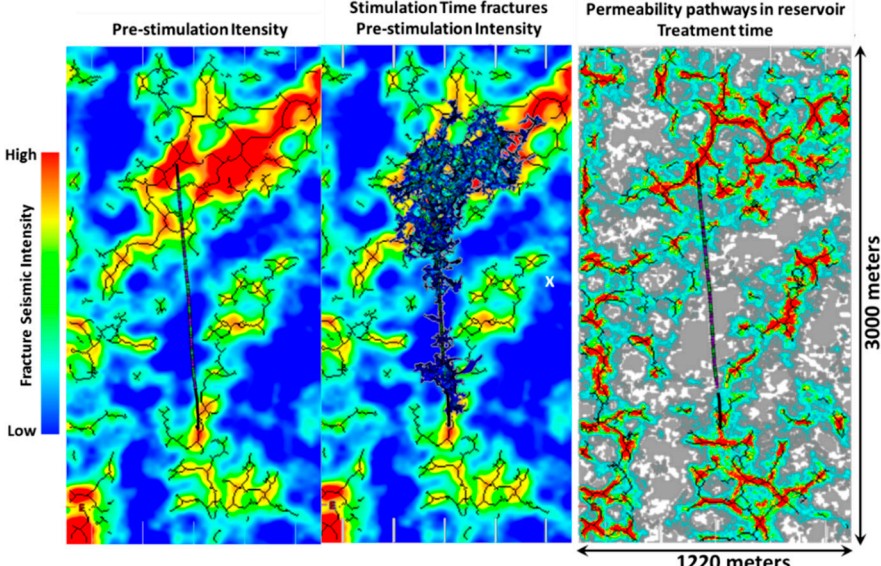

**Figure 29.** The fracture seismic intensity volume computed from fracture seismic data recorded before stimulation show that the stimulation activated the same fractures that were mapped before the stimulation activity. The left panel shows a depth slice of the intensity volume from the pre-stimulation data. The middle panel shows the depth slice of the pre-stimulation intensity with the overlay of the fractures active during stimulation. The right panel shows the connectivity pathways computed from the 16 intensity volumes computed for each stage.

### 3.7. Pump Startup Time Fracture System—Texas

This example shows how the pressure from the stimulation produces fracture seismic intensity in the fracture system before formation breakdown. Formation breakdown is when the fractures near the well open and allow the initiation of fluid flow into the reservoir. The fracture seismic data recorded before formation breakdown can provide very useful details about the reservoir.

Figure 30 shows a spectrogram for 30 min of data recorded in the Eagle Ford shale during pressure buildup for the first stage of a hydraulic stimulation. During startup, the increasing pressure moves into the rocks causing resonances in the permeable fractures that are connected to the well at Stage 1. The observed resonances grow in amplitude and complexity with increasing pressure. The resonances transition from low-amplitude dispersive to high-amplitude turbulent after formation breakdown. These resonances are very different from those observed during the pre-stimulation in the New Albany shown in Figure 28.

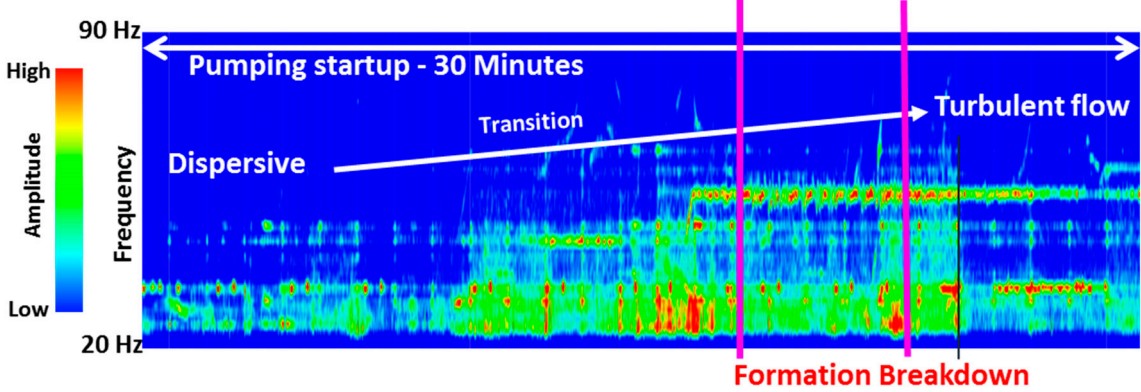

**Figure 30.** Spectrogram during Stage 1 startup shows the resonances transition from low amplitude to high amplitude as the pressure is increased with pumping time. Formation breakdown causes a substantial change in the resonance as the fluid begins to flow into the formation. The lavender bars mark the 5 min of data that are used to compute the fracture images.

The 5 min of trace data marked by the lavender lines in Figure 30 were selected for computation of the pre-breakdown fracture seismic volume. These minutes were selected because of the high fracture seismic signals in the spectrogram and because it is before formation breakdown. Previously, it was not expected that the fractures would emit such high-intensity fracture seismic resonances before formation breakdown.

The depth slice at the well depth of the fracture seismic intensity volume computed from these 5 min is shown in Figure 31. The feature trending from SW to NE and terminating at the well is a fault that was previously observed in the 3D reflection seismic data. The depth slice shows the fracture seismic intensity stimulated by the startup of pumping with the overlay in black lines of the fractures. The fractures map the connectivity to the perforation location for Stage 1.

This section of the Eagle Ford has a large number of fractures, as can be seen in the 3D reflection section shown in Figure 23. The interpretation is that only a few of these fractures were activated during Stage 1 startup. Most of the fractures in the volume are not activated with the increase in pressure. Only the most permeable fractures are activated during the startup time before the first stage formation breakdown. The fracture lines form a pattern that might have been in related to a previous stress direction that is different from the one in place today.

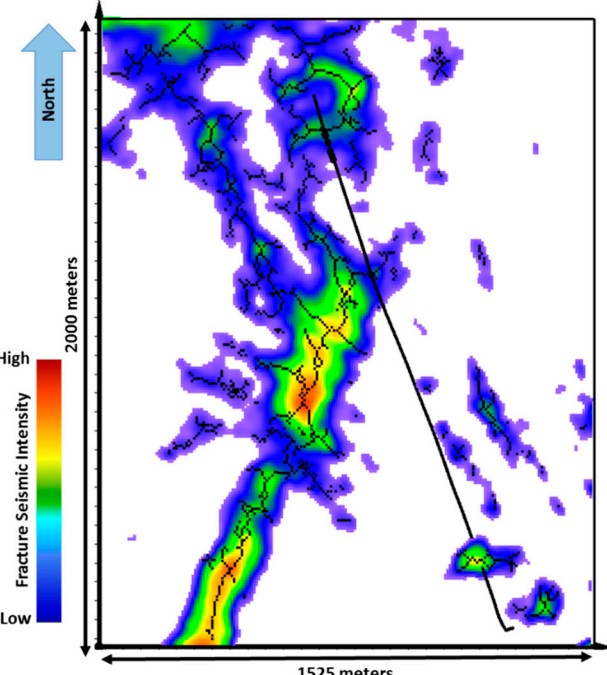

**Figure 31.** Depth slice of fracture intensity volume computed for 5 min fracture seismic data and the overlay of the computed fractures. The fractures show the connectivity pathways that connect to the well at the Stage 1 perforations.

### 3.8. Stimulation Time—Texas

The fracture seismic fracture surfaces shown in Figure 32 are computed from data recorded during the stimulation for a well in the Permian. The fractures activated by the stimulation open out into the reservoir for approximately 15 m and then turn parallel to the well. This well was not economic because the fractures that opened did not have sufficient rock volume. The interpretation is that the well was either not drilled along the maximum horizontal stress direction or that the pumping pressures changed the local stress causing the fractures to turn parallel to the well.

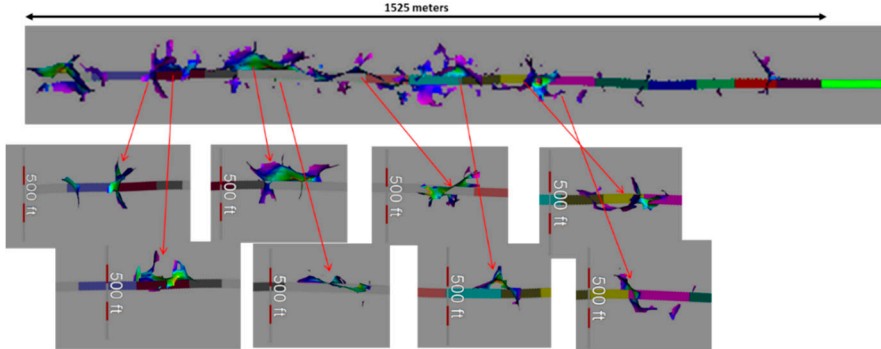

**Figure 32.** Fracture images computed during stimulation for the stages of a well in the Permian showing the that the stimulated fractures open perpendicular to the well and then turn parallel to the well.

### 3.9. Prediction of Well Interferences on Adjacent Well—Pennsylvania

The location accuracy of fracture seismic fracture imaging is demonstrated in a project in the Marcellus. One well had been drilled and seven more planned when a buried grid was installed over the pad site. Well A in Figure 33 was stimulated and put on production before Well B was drilled. The fracture seismic signals initiated by the fluid flow into Well A were used to compute a fracture seismic intensity volume before Well B was stimulated.

Well A was in production during this fracture seismic recording and the fractures computed from the fracture seismic data intersect with the path for Well B. This indicates that when Well B is fracked, there could be pressure hits on Well A at three separate stages of the Well B stimulation. The locations of these three predictions are shown by the circles in Figure 33. Later, when Well B was stimulated, pressure changes in Well A were recorded by a gauge at the head of Well A. These pressure changes occurred for treatments located at fracture seismic imaged fracture crossings. These engineering data confirm the predicted connection between the wells and that the fracture map shows the fractures that transmitted the pressure from Well B to Well A.

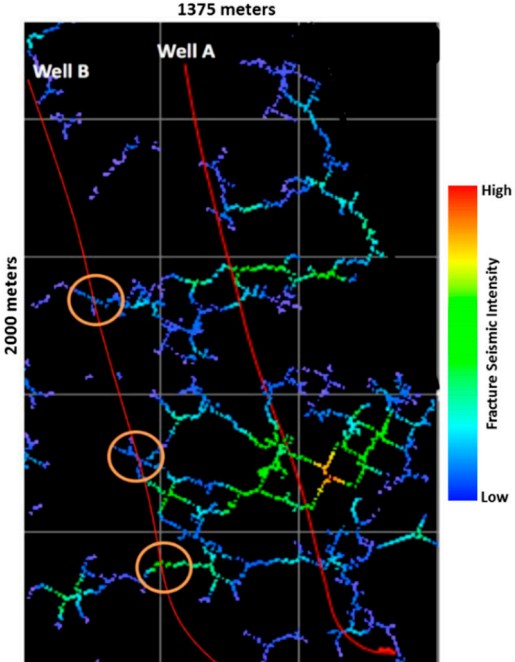

**Figure 33.** Predicting pressure interferences on an adjacent well. The fracture intensity computed before the stimulation of Well B but while Well A was producing show the fractures crossing the path of Well B. The locations shown by the circles mark the stage locations along Well B that were being stimulated when pressure interferences were measured at the well head of Well A.

### 3.10. Actively Producing Volume Before and After Pressure Hits—Pennsylvania

The fluid flow into the well during production causes turbulent resonance in the fractures and allows for the computation of the actively producing volumes. Fracture seismic intensity volumes computed from data recorded over a producing well are used to extract the active voxels that are connected to the well. An intensity threshold is first selected and applied to the intensity volume. The remaining voxels that are connected to the well are extracted from the fracture seismic intensity volume using an iterative process whereby the active voxels that are touching the well are extracted in the first iteration. The subsequent iterations detect and extract voxels that are touching the previously extracted voxels. The iterations continue until no more active voxels connected to the well are detected. The volume of extracted voxels is the actively producing volume that can be used for planning additional wells or reservoir treatments.

Two wells were drilled in this example from the Marcellus Shale shown in Figure 34. As discussed for Figure 33, fracture seismic was recorded before Well B was stimulated but while Well A was in production. The producing volume for Well A was computed using fracture seismic recorded before the stimulation of Well B (Figure 34, left) and again after the stimulation of Well B (Figure 34, right). Comparing the two producing volumes shows that the producing volume for Well A is 25% smaller after the stimulation of Well B than it was before the stimulation.

The production data recorded at the wellhead of Well A show that the production from Well A was reduced by 30% during the stimulation of Well B. The well head production reduction agrees with the fracture seismic intensity volume reduction and supports the interpretation that the hits on Well A caused the reduction in production.

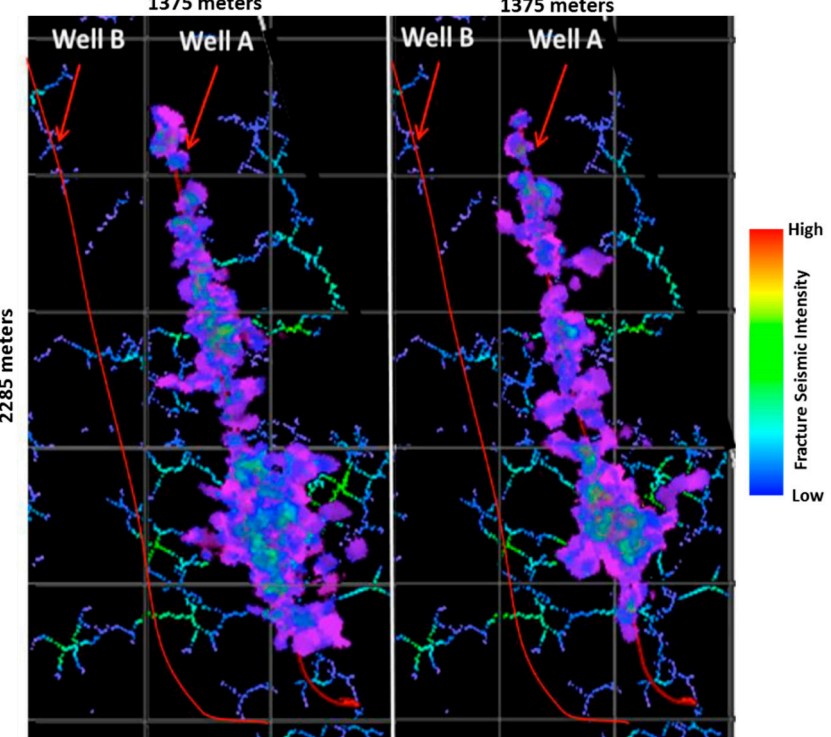

**Figure 34.** Producing rock volume around Well A before (left) and after (right) the stimulation of Well B. The pressure changes in Well A (Figure 33) during the stimulation of Well B caused production declines and also reduction in the producing volume.

### 3.11. Fluid Producing Volumes over Time—Texas

The Eagle Ford producing volume images on the left in Figure 35 show the fracture seismic intensity volumes computed using fracture seismic recorded on a permanent buried grid for the

stimulation time, at two years and at three years after the well was put on production. The buried grid was activated during the stimulation and again two years and three years later.

Computing the fracture seismic intensity volumes for each of these times allows the active voxels connected to the well to be extracted for each time. The top down view of the stimulated rock volume and the actively producing volumes are shown. These three time-lapse volumes show that the stimulated volume during the treatment is much larger than the producing volume after two and three years of production. After two years, there are portions of the well that are not producing and after three years, only short segments of the well are producing.

The production curves in the right panel of Figure 35 show the volume of production measured at the well head. Zooming into the curves for the times of the active producing volumes shows that the well head production volumes correlate with the active volumes measured from the fracture seismic monitoring. The fracture seismic recording time was at a time when the well was producing and the fracture seismic actively producing volume is large. The recording at year three was at a time when the production from this well was very low and perhaps shut in and the fracture seismic actively producing volume is small.

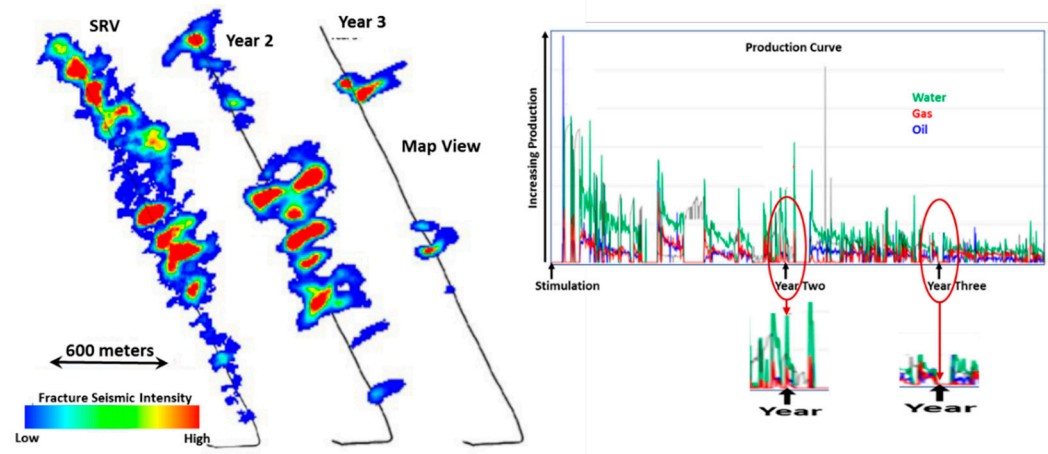

**Figure 35.** Fracture seismic intensity signals decline over three years. Left: Fracture seismic intensity volumes showing the rock volume activated during stimulation, the rock volume that is active after two years of production, and the rock volume that is active after three years of production. Right: Production curves recorded at the well head showing the fluids produced over time. The passive data for year three were recorded at a time when the production curves show the well was not in production.

### 3.12. Forecasting Production Before Drilling—Texas

This example shows an attempt to extract the reservoir connectivity volume that is connected to a well path before the well is drilled. Recording using a buried grid allowed extraction of the intensity along the planned well path before drilling and again during production 2.5 years after it was put on production. The fracture seismic intensity volumes shown in Figure 36 compare the predicted producing volume before the well was drilled to the actively producing volume after 2.5 years of production. The data were recorded twice using a buried grid. The first recording was two months before the well was drilled and the second recording was 2.5 years after the well was put on production. The planned well path for this well was used for the pre-drill active voxel extraction from the intensity volume. Figure 36 shows the forecast producing volume in black (left) and the measured producing volume 2.5 years later in blue (middle). Their overlay is shown on the right. There is good correlation between the forecast and the measured producing volumes.

These fracture seismic intensity volumes show that the production is coming from zones in the reservoir that were permeable before the well was drilled. This result establishes that time-lapse monitoring of the reservoir using fracture seismic, from pre-development through the production life of the reservoir, provides essential information for optimal management of the reservoir.

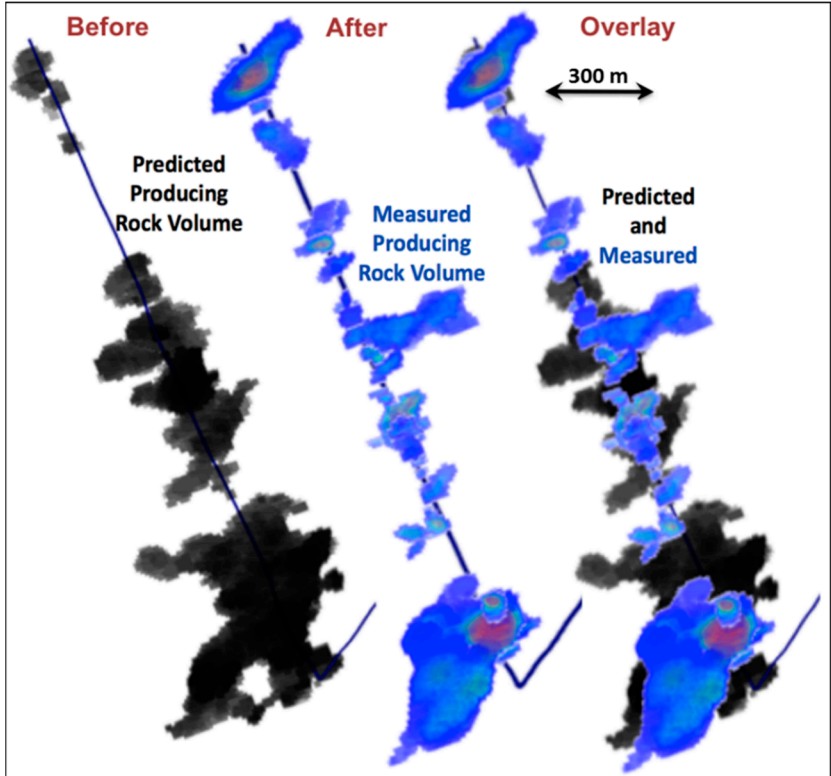

**Figure 36.** Forecasting production before the well is drilled. Left side: Pre-drill, fracture seismic intensity-based forecast of the producing rock volume around the well. Center: Fracture seismic-based measurement of the producing volume after 2.5 years of production. Right: Overlay of the forecast and the observed producing rock volumes.

## 4. Conclusions

Based on several converging lines of evidence, we and others have concluded that fracture seismic recordings contain episodic signals generated by resonating, fluid-filled fractures. Theoretical studies by others show these resonances can be generated by interfering seismic waves originating from either dislocation at the fracture tips or turbulent fluid entries from other fractures and hydraulic stimulations. In this paper, we focus on showing how these signals can be used to map the most permeable structures in the subsurface.

Given their long durations and narrow-band frequency content, these signals can be understood as harmonic vibrations of fluid-filled cracks that are elastically coupled to the surrounding rock mass. The method for recording, processing, and imaging these signals is called fracture seismic in order to distinguish it from micro-seismic methods. Micro-seismic methods detect the impulsive dislocations that generate distinguishable P and S waves to locate fractures. Fracture seismic captures and images the signals from the entire fracture and builds a three-dimensional image of the fractures. Fractures that are interpreted from micro-seismic have only a partial correlation with the fracture systems mapped using the fracture seismic method.

Fracture seismic observations can be acquired with both high density, multi-receiver, reflection seismographic equipment and lower density buried grids. The number of receivers necessary per square km of study area is between 30 and 60 for surface-based observations and 1 to 3 for buried grids. The density of the receiver grid for any acquisition is determined by the noise environment for the project and the economics of permitting and physical access constraints. The quality of the fracture seismic map is impacted by the density. The area to be instrumented is determined by both the depth and area to be imaged. From the edge of the area to be imaged, the farthest offset receivers should be 1.2 to 1.5 times the depth of the target area.

The fracture seismic method computes fracture emission intensity volumes using one-way depth migration. These intensity volumes can be computed using modern digital signal processing

of fracture seismic data recorded during the acquisition of multi-receiver reflection seismic survey data. The redundancy of such data allows for the removal of other passively recorded signals, including earthquakes and cultural and industrial generated background noise. Well-known seismic reflection processing codes such as cepstral filters, noise analysis and filtering, and depth migration, can be readily adapted to the one-way-travel-time depth migration used in the fracture seismic method for computing fracture seismic intensity.

A substantial base of fracture seismic observational case histories now exists. These examples establish that fracture seismic methods can reveal the locations of the subsurface fluid-flow pathways. Pre-drill fracture seismic mapping can be used to guide well paths, establish optimal treatment programs, and forecast well interferences. Stimulation time fracture seismic can be used to measure treatment effectiveness. Combined with pre-treatment fracture seismic maps, both potential and actual fluid production can be readily and accurately estimated. Time-lapse fracture seismic tracks the evolution of flow paths over time.

These attributes of fracture seismic permeable structure mapping establish its importance in future exploration, development, production, and management of subsurface resources. With the rapid expansion of the number of receivers that can be fielded and the speed of modern computers, fracture seismic acquisition can be integrated with 3D seismic reflection acquisition. Both reflection seismic volumes for detailed interpretation of the geologic structure and fractures seismic intensity volumes can be computed simultaneously and allow the integration of the subsurface connectivity with the geologic formations. As a consequence of these developments and the value of results of our case studies, we believe that studies of the kind we have presented here will soon become standard practices, for both commercial and social purposes.

**Author Contributions:** Conceptualization, C.S.; formal analysis, C.S. and P.M.; investigation, C.S. and P.M.; methodology, C.S.; project administration, C.S.; software, C.S.; supervision, C.S.; validation, P.M.; writing—original draft, C.S. and P.M.; writing—review and editing, C.S. and P.M..

**Funding:** This research received no external funding

**Acknowledgments:** The authors wish to thank Jan Vermilye for her ideas and contributions to many of the methods presented and for her review of the manuscript. Ashley Yaner, Amanda Klaus, and Lance Bjerke processed the data and provided the data integration in the field examples.

**Conflicts of Interest:** The authors declare no conflict of interest.

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
