# Peer review of "Fracture Seismic: Mapping Subsurface Connectivity"

_geosciences, doi:10.3390/geosciences9120508_

Round 1

Reviewer 1 Report

Comments on: ‘Fracture Seismic: Mapping subsurface connectivity’ by Charles Sicking, and Peter Malin.

I have read the manuscript, and I found it to be an excellent work which is appropriate to the scope of this journal as the subject/topic is of enough interest that deserves to be addressed to the scientific community working in this field.

In my opinion, the manuscript is well written, the sections are rightly supported by impressive graphics and data, and in general, this manuscript is a clear example about what it is expected from a research paper.

In my opinion, is an extraordinary example of a well-done work about fracture seismic, in which not only deals with an important and interesting topic for the scientific community but also, the writing and its content have been undertaken perfectly. I mean, explain properly the state-of-art of this issue, introduce the background and the methodology, data to finish, finally, with the description of some case studies about this issue. So for me, those are some of the strong points about this manuscript.

So, I would suggest the paper to be accepted in its present form.

Author Response

Authors reply to reviewer number 1:

I have read your comments and appreciate you review.  Thank you for you efforts. 

I do not have specific comments other than to say that I appreciate your positive response to the manuscript. 

Thank you

Dr. Charles Sicking

Reviewer 2 Report

This an interesting article on theoretical and practical  possibilities of fracture seismic. It is a well-organized with sufficiently clear theoretical basis and many of application examples. It is wrote a good highly specialized language. It is also good illustrated, but the quality of many figures could be better.

Considering technical side of the manuscript I suggest to consider:

1) Could you try to precise in the introductory part, who the first used a term “fracture seismic”.

2) It will be interested to write in “acquisition” chapter about receivers used, particularly about its frequency band.

3) Could you write more clear, why the fracture seismic signals are in the lowest 2% of the quefrencies (lines 353-358).

4) Keywords: “ambient seismic; passive” seem to be not properly used.

From editorial side of the manuscript, I suggest to improve:

1) There are many figures poor informative.  Many figures do not have axis description e.g. 5, 10, 25, 29. Please, indicate clearly on many figures, what kind of section is presented – horizontal or vertical. In many cases, you should clearly described what is, for example, on a left and a right hand side part of the figure (e.g. fig. 20). Indicate high amplitude on figure 22 (right hand side). Caption of figure 33 should be in italic.

2) You should consequently use metric system (SI) in the text and in many of figures.

3) It seems that form of citation used is not correct e.g. (Kauklis, 1962)[1]. Change also the name on Krauklis.

Author Response

Response to reviewer number 2:

comment #1:  I have added some words concerning the definition and first use of "Fracture Seismic Method".  This did need some additional explaination.

Comment #2:  I have provided a description of the surface geophones used in recording passive seismic for the purpose of imaging fractures using the Fracture Seismic Method. 

Comment #3:  Why are the signals in the lowest 2% of the quefrency domain?  I edited the description of Cepstral filtering to clarify this. 

Comment #4:  I changed the list of keywords to make them more fitting to this paper. 

For the editorial side of the paper:

Point #1:  I edited many of the figures to add axes where they were missing, labeled the amplitudes where appropriate, and other edits.  More than 50% of the figures have been modified and replaced in the Word document.  The High resolution figures must now be re-uploaded to the site. 

Point #2:  All of the units used in the manuscript are now in SI (metric) units.  I searched the text and the figures to find all of the English units and replace them with metric. 

Point 3#:  I did a spell check and corrected the spellings that I found.  The form of the references in the text are as close to that stated in the documents that were provided.  If there are problems with the format of the references in the text, I will happily change them to fit the requirements.  I spent a lot of effort to meet your requirements in the original submission.  An example that needs to be changed and how to change it would be most helpful. 

Thank you for your detailed review.  It is most appreciated. 

Dr Charles Sicking